# SigDiffusions: Score-Based Diffusion Models for Time Series via Log-Signature Embeddings

**Barbora Barancikova**
Department of Computing
Imperial College London

**Zhuoyue Huang**
Department of Mathematics
Imperial College London

**Cristopher Salvi**
Department of Mathematics
Imperial College London

## Abstract

Score-based diffusion models have recently emerged as state-of-the-art generative models for a variety of data modalities. Nonetheless, it remains unclear how to adapt these models to generate long multivariate time series. Viewing a time series as the discretisation of an underlying continuous process, we introduce `SigDiffusion`, a novel diffusion model operating on log-signature embeddings of the data. The forward and backward processes gradually perturb and denoise log-signatures while preserving their algebraic structure. To recover a signal from its log-signature, we provide new closed-form inversion formulae expressing the coefficients obtained by expanding the signal in a given basis (e.g. Fourier or orthogonal polynomials) as explicit polynomial functions of the log-signature. Finally, we show that combining `SigDiffusions` with these inversion formulae results in high-quality long time series generation, competitive with the current state-of-the-art on various datasets of synthetic and real-world examples.

## 1 Introduction

Time series generation has gained significant attention in recent years due to the growing demand for high-quality data augmentation in fields such as healthcare (Trottet et al., 2023) and finance (Hwang et al., 2023). Since sampling rates are often arbitrary and non-uniform, it is natural to assume that the data is collected from measurements of some underlying physical system evolving in continuous time. This perspective calls for modelling tools capable of processing temporal signals as continuous functions of time. We will often refer to such functions as *paths*.

The idea of representing a path via its iterated integrals has been the object of numerous mathematical studies, including geometry (Chen, 1957; 1958) control theory (Fliess et al., 1983), and stochastic analysis (Lyons, 1998). The collection of these iterated integrals is known as the *signature* of a path. Thanks to its rich algebraic and analytic properties summarised in Section 2, the signature is an efficient-to-compute, universal feature map for time series evolving in continuous time. As a result, signature methods have recently become mainstream in many areas of machine learning dealing with irregular time series, from deep learning (Kidger et al., 2019; Morrill et al., 2021; Cirone et al., 2023; 2024) to kernel methods (Salvi et al., 2021a; Lemercier et al., 2021b; Issa et al., 2024), with applications in quantitative finance (Arribas et al., 2020; Salvi et al., 2021b; Horvath et al., 2023; Pannier & Salvi, 2024), cybersecurity (Cochrane et al., 2021), weather forecasting (Lemercier et al., 2021a), and causal inference (Manten et al., 2024). For a concise summary of this topic, we refer the interested reader to Fermanian et al. (2023b).

Score-based diffusion models have emerged as a powerful framework for modelling complex distributions in computer vision, audio, and text (Song et al., 2020; Biloš et al., 2023; Popov et al., 2021; Cai et al., 2020; Voleti et al., 2022). These models perturb the observed data distribution through a forward diffusion process, adding noise until an easy-to-sample base distribution is reached. A new sample from the learned data distribution is then generated by reversing the noising process using a learned *score* function of the data conditional on the noise level. Despite recent efforts summarised in Section 4, it remains unclear how to adapt score-based diffusion models for generating long signals in continuous time.

**Contributions** In this paper, we make use of the *log-signature*, a compressed version of the signature, as a parameter-free Lie algebra embedding for time series. In Section 2.3, we introduce `SigDiffusion`, a new diffusion model that gradually perturbs and denoises log-signatures preserving their algebraic structure. To recover a path from its log-signature embedding, we provide novel closed-form inversion formulae in Section 3. Notably, we prove that the coefficients of the expansion of a path in a given basis, such as Fourier or orthogonal polynomials, can be expressed as explicit polynomial functions on the log-signature. Our results provide a major improvement over existing signature inversion algorithms (see Section 5.1) which suffer from scalability issues and are only effective on simple examples of short piecewise-linear paths. In Section 5, we demonstrate how the combination of `SigDiffusions` with our inversion formulae provides a time series generative approach, competitive with state-of-the-art diffusion models for temporal data on various datasets of synthetic and real-world examples.

## 2 GENERATING LOG-SIGNATURES WITH SCORE-BASED DIFFUSION MODELS

We begin this section by recalling the relevant background material before introducing our `SigDiffusion` model. We will limit ourselves to reporting only the key properties of signatures and the notation necessary for the inversion formulae in Section 3. Additional examples of signature computations can be found in Appendix A.3.

### 2.1 THE (LOG)SIGNATURE

Let $x : [0,1] \to \mathbb{R}^d$ be a smooth $d$-dimensional time series defined on a time interval $[0,1]$. We will equivalently refer to this object as a *path*. The *step-$n$ signature* $S^{\leq n}(x)$ of $x$ is defined as the following collection of iterated integrals

$$S^{\leq n}(x) = (1, S_1(x), \ldots, S_n(x)) \tag{1}$$

where

$$S_k(x) = \int_{0 \leq t_1 < \ldots < t_k \leq 1} dx_{t_1} \otimes \ldots \otimes dx_{t_k} \quad \text{for } 1 \leq k \leq n$$

and $\otimes$ denotes the tensor product. Intuitively, one can view the signature as a set of tensors of increasing dimension, where the value of the $m$-th tensor at the index $i_1, i_2, \ldots, i_m$ represents a measure of "interaction" between the $i_1, i_2, \ldots, i_m$-th channels of $x$. This makes the signature transform particularly effective at capturing information about the shape of multivariate time series.

**Example 2.1.** *Assume $d = 2$, and denote the two channels of $x$ as $x = (x^1, x^2)$. Then $S_1(x), S_2(x)$ are tensors with shape $[2]$ and $[2,2]$ respectively*

$$S_1(x) = \int_0^1 dx_{t_1} = \begin{pmatrix} \int_0^1 dx_{t_1}^1 \\ \int_0^1 dx_{t_1}^2 \end{pmatrix},$$

$$S_2(x) = \int_0^1 \int_0^{t_2} dx_{t_1} \otimes dx_{t_2} = \begin{pmatrix} \int_0^1 \int_0^{t_2} dx_{t_1}^1 dx_{t_2}^1 & \int_0^1 \int_0^{t_2} dx_{t_1}^2 dx_{t_2}^1 \\ \int_0^1 \int_0^{t_2} dx_{t_1}^1 dx_{t_2}^2 & \int_0^1 \int_0^{t_2} dx_{t_1}^2 dx_{t_2}^2 \end{pmatrix}.$$

Denoting the standard basis of $\mathbb{R}^d$ as $e_1, e_2, \ldots, e_d$ we define a basis of the space of $k$-dimensional tensors as

$$e_{i_1 i_2 \ldots i_k} = e_{i_1} \otimes e_{i_2} \otimes \ldots \otimes e_{i_k}, \quad \text{for} \quad 1 \leq i_1, \ldots, i_k \leq d \text{ and } 0 \leq k \leq n.$$

We refer to these basis elements as *words*. In Section 3, we will make use of the notation $\langle e_{i_1 i_2 \ldots i_k}, S^{\leq n}(x) \rangle \in \mathbb{R}$ to extract the $(i_1, \ldots, i_k)^{th}$ element of the $k$-th signature tensor $S_k(x)$.

Words can be manipulated by two key operations: the *shuffle product* ⧢ and *right half-shuffle product* ≻. The shuffle product of two words of length $r$ and $s$ (with $r + s \leq n$) is defined as the sum over the $\binom{r+s}{s}$ ways of interleaving the two words. For a formal definition, we refer readers to Reutenauer (2003, Section 1.4). Much of the internal structure of the signature is characterized by the *shuffle identity* (see Lemma A.0.1), which uses the *shuffle* and *half-shuffle products* to describe the relationship between elements of higher and lower-order signature tensors. This identity is crucial

in our derivation of the inversion formulae in Appendix C. A rigorous algebraic explanation of these concepts is provided in Appendix A.1.

Moreover, it turns out that the space of signatures has the structure of a *step-$n$ free nilpotent Lie group $\mathcal{G}^n(\mathbb{R}^d)$*. We denote by $\mathcal{L}^n(\mathbb{R}^d)$ the unique Lie algebra associated with $\mathcal{G}^n(\mathbb{R}^d)$, and we call its elements *log-signatures*. $\mathcal{G}^n(\mathbb{R}^d)$ is the image of the $\mathcal{L}^n(\mathbb{R}^d)$ under the exponential map

$$\mathcal{G}^n(\mathbb{R}^d) = \exp(\mathcal{L}^n(\mathbb{R}^d)) \tag{2}$$

where, in the case of signatures, $\exp$ denotes the tensor exponential defined in Appendix A.2. Furthermore, one can use the tensor logarithm (see Equation (13)) to convert log-signatures to signatures. These two operations are mutually inverse.

We note that the Lie algebra $\mathcal{L}^n(\mathbb{R}^d)$ is a vector space of dimension $\beta(d, n)$ with

$$\beta(d, n) = \sum_{k=1}^{n} \frac{1}{k} \sum_{i|k} \mu\left(\frac{k}{i}\right) d^i,$$

where $\mu$ is the Möbius function (Reutenauer, 2003). Crucially, the Lie algebra is isomorphic to the Euclidean space $\mathbb{R}^{\beta(d,n)}$, which motivates the diffusion model architecture in Section 2.3.

## 2.2 SIGNATURE AS A TIME SERIES EMBEDDING

The (log)signature exhibits additional properties making it an especially interesting object in the context of generative modelling for sequential data. In this section we summarise such properties without providing technical details, as these have been discussed at length in various texts in the literature. For a thorough review, we refer the interested reader to Cass & Salvi (2024, Chapter 1).

At first glance, computing the (log)signature seems intractable for general time series. However, an elegant and **efficient computation** is possible due to *Chen's relation*

**Lemma 2.0.1** (Chen's relation). *For any two smooth paths $x, y : [0, 1] \rightarrow \mathbb{R}^d$ the following holds*

$$S^{\leq n}(x * y) = S^{\leq n}(x) \cdot S^{\leq n}(y), \tag{3}$$

*where $*$ denotes path-concatenation, and $\cdot$ is the signature tensor product defined in Equation (11).*

Combining Chen's relation with the fact that the signature of a linear path is simply the tensor exponential of its increment (see Example A.2) provides us with an efficient algorithm for computing signatures of piecewise linear paths. This approach eliminates the need to calculate integrals when computing signature embeddings. See Appendix A.3 for simple examples of computations.

Furthermore, the (log)signature is *invariant under reparameterisations*. This property essentially allows the signature transform to act as a filter that removes an infinite dimensional group of symmetries given by time reparameterisations. The action of reparameterising a path can be thought of as sampling its observations at a different frequency, resulting in **robustness to irregular sampling**. Another important property of the signature is the **factorial decay in the magnitude of its coefficients** (Cass & Salvi, 2024, Proposition 1.2.3). This fast decay implies that truncating the signature at a sufficiently high level retains the bulk of the critical information about the underlying path.

The signature is *unique* for certain classes of paths, ensuring **a one-to-one identifiability with the underlying path**. An example of such classes is given by paths which share an identical, strictly monotone coordinate and are started at the same origin. More general examples are discussed in Cass & Salvi (2024, Section 4.1). This property is important if one is interested, as we are, in recovering the path from its signature. Yet, providing a viable algorithm for inverting the signature has, until now, been challenging; valid but non-scalable solutions have been proposed only for special classes of piecewise linear paths (Chang & Lyons, 2019; Fermanian et al., 2023a; Kidger et al., 2019). In Section 3 we provide new closed-form inversion formulae that address this limitation.

## 2.3 DIFFUSION MODELS ON LOG-SIGNATURE EMBEDDINGS

As described in Section 2.1, any element of $\mathcal{G}^n(\mathbb{R}^d)$ corresponds to the step-$n$ signature of a smooth path. Taking the tensor logarithm in Equation (13) then implies that an arbitrary element of $\mathcal{L}^n(\mathbb{R}^d)$

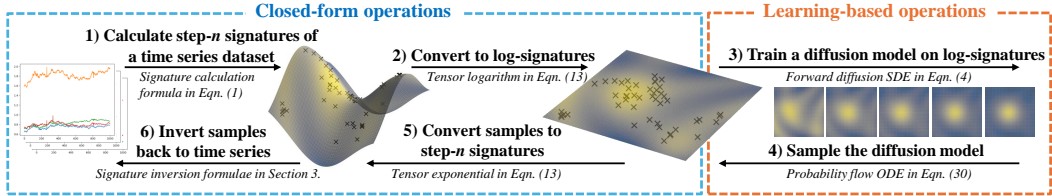

Figure 1: `SigDiffusions` pipeline. The signatures of a time series dataset are points distributed in a non-Euclidean space (Lie group). Converting to log-signatures maps them to a Euclidean space (Lie algebra) where standard diffusion models operate. Calculating the log-signature embedding and its inverse (blue box) are fully deterministic operations, which greatly simplifies the learning task. The log-signatures serve as inputs to a score-based diffusion model (orange box). Step 6 is enabled by our newly derived closed-form inversion formulae.

corresponds to the *step-$n$ log-signature* of a smooth path. Because the Lie algebra $\mathcal{L}^n(\mathbb{R}^d)$ is a linear space, adding two log-signatures yields another log-signature. Furthermore, the dimensionality $\beta(d, n)$ of $\mathcal{L}^n(\mathbb{R}^d)$ is strictly smaller than $\frac{d^{n+1}-1}{d-1}$, making the log-signature a more compact representation of a path than the signature while retaining the same information. We can leverage these two properties to run score-based diffusion models on $\mathcal{L}^n(\mathbb{R}^d)$, followed by an explicit signature inversion, which we discuss in the next section. Figure 1 presents an overview of the proposed idea, which we demonstrate experimentally in Section 5 to be an efficient and high-quality method for generating multivariate time series.

We briefly recall that score-based diffusion models progressively corrupt data with noise until it reaches a tractable form, then learn to reverse this process to generate new samples from the underlying data distribution $p(\mathbf{x})$. A neural network is used to estimate the gradient of the log probability density, known as the *score* (Song & Ermon, 2019), $s_\theta(t, \mathbf{x}) \approx \nabla_\mathbf{x} \log p_t(\mathbf{x})$ at each injected noise level $t$. We model the forward data perturbation process with a stochastic differential equation (SDE) of the form

$$d\mathbf{x} = -\frac{1}{2}\beta(t)\mathbf{x}dt + \sqrt{\beta(t)}d\mathbf{w}, \; \mathbf{x}(0) \sim \text{data} \tag{4}$$

where $\beta(t)$ is linear over $t \in [0, 1]$. The corresponding reverse diffusion process follows a reverse-time SDE (Anderson, 1982) $d\mathbf{x} = \left[-\frac{1}{2}\beta(t)\mathbf{x} - \beta(t)\nabla_\mathbf{x} \log p_t(\mathbf{x})\right] dt + \sqrt{\beta(t)}d\overline{\mathbf{w}}$, where $t$ flows backwards from $T$ to $0$ and $\overline{\mathbf{w}}$ is Brownian motion with a negative time step $dt$. The initial point $\mathbf{x}(T)$ is sampled from a standard Gaussian distribution. Following previous work (Ho et al., 2020; Biloš et al., 2023; Yuan & Qiao, 2024), we use a simple transformer architecture with sinusoidal positional embeddings of $t$ to model $s_\theta$.

## 3 SIGNATURE INVERSION

In this section, we provide explicit signature inversion formulae. We do so by expressing the coefficients of the expansion of a path in the Fourier or orthogonal polynomial bases as a linear function on the signature. The necessary background material on orthogonal polynomials and Fourier series can be found in Appendix B.

Throughout this section, $x : [0, 1] \to \mathbb{R}$ will denote a 1-dimensional smooth path. The results in the sequel can be naturally extended to multidimensional paths by applying the same procedure channel by channel. Depending on the type of basis we chose to represent the path, we will often need to reparameterise the path from the interval $[0, 1]$ to a specified time interval $[a, b]$ and augment it with time as well as with additional channels $c^1, c^2, ..., c^r : [a, b] \to \mathbb{R}$, tailor-made for the specific type of inversion. We denote the augmented path by $\hat{x}(t) = (t, c^1(t), ..., c^r(t), x(t)) \in \mathbb{R}^{r+2}$. Note that these transformations are fully deterministic and do not affect the complexity of the generation task outlined in Section 2.3. Furthermore, we will use the shorthand notation $S(\hat{x})$ for the step-$n$ signature $S^{\leq n}(\hat{x})$ throughout the section, and assume that the truncation level $n$ is always high enough to retrieve the desired number of basis coefficients. All proofs can be found in Appendix C.

## 3.1 INVERSION VIA FOURIER COEFFICIENTS

In this section, we derive closed-form expressions for retrieving the first $n$ Fourier coefficients of a path from its signature. First, recall that the Fourier series of a $2\pi$-periodic path $x(t)$ up to order $n \in \mathbb{N}$ is $x_n(t) = a_0 + \sum_{n=1}^{n}(a_n \cos(nt) + b_n \sin(nt))$ where $a_0, a_n, b_n$ are defined as

$$a_0 = \frac{1}{2\pi} \int_0^{2\pi} x(t)dt, \tag{5}$$

$$a_n = \frac{1}{\pi} \int_0^{2\pi} x(t)\cos(nt)dt, \tag{6}$$

$$b_n = \frac{1}{\pi} \int_0^{2\pi} x(t)\sin(nt)dt. \tag{7}$$

**Theorem 3.1.** *Let $x : [0, 2\pi] \to \mathbb{R}$ be a periodic smooth path such that $x(0) = 0$, and consider the augmentation $\hat{x}(t) = (t, \sin(t), \cos(t) - 1, x(t)) \in \mathbb{R}^4$. Then the following relations hold*

$$a_0 = \frac{1}{2\pi}\langle e_4 \succ e_1, S(\hat{x})\rangle,$$

$$a_n = \frac{1}{\pi}\sum_{k=0}^{n}\sum_{q=0}^{k}\binom{n}{k}\binom{k}{q}\cos(\frac{1}{2}(n-k)\pi)\langle(e_4 \shuffle e_2^{\shuffle n-k} \shuffle e_3^{\shuffle q}) \succ e_1, S(\hat{x})\rangle,$$

$$b_n = \frac{1}{\pi}\sum_{k=0}^{n}\sum_{q=0}^{k}\binom{n}{k}\binom{k}{q}\sin(\frac{1}{2}(n-k)\pi)\langle(e_4 \shuffle e_2^{\shuffle n-k} \shuffle e_3^{\shuffle q}) \succ e_1, S(\hat{x})\rangle. \tag{8}$$

## 3.2 INVERSION VIA ORTHOGONAL POLYNOMIALS

To accommodate path generation use cases for which a non-Fourier representation is more suitable, next we derive formulae for inverting the signature using expansions of the path in orthogonal polynomial bases. Recall that any orthogonal polynomial family $(p_n)_{n\in\mathbb{N}}$ with a weight function $\omega : [a, b] \to \mathbb{R}$ satisfies a three-term recurrence relation

$$p_n(t) = (A_n t + B_n)p_{n-1}(t) + C_n p_{n-2}(t), \qquad n \geq 2, \tag{9}$$

with $p_0(t) = 1$ and $p_1(t) = A_1 t + B_1$. Also, note that any smooth (or at least square-integrable) path $x(t)$ with $x(a) = 0$ can be approximated arbitrarily well as $x(t) \approx \sum_{n=0}^{\infty} \alpha_n p_n(t)$ where $\alpha_n$ is the $n$-th orthogonal polynomial coefficient

$$\alpha_n = \frac{1}{(p_n, p_n)} \int_a^b x(t)p_n(t)\omega(t)dt, \tag{10}$$

and $(\cdot, \cdot)$ denotes the inner product $(f, g) = \int_a^b f(t)g(t)\omega(t)dt$. We include several examples of such polynomial families in Appendix B.

**Theorem 3.2.** *Let $x : [a, b] \to \mathbb{R}$ be a smooth path such that $x(a) = 0$. Consider the augmentation $\hat{x}(t) = (t, \omega(t)x(t)) \in \mathbb{R}^2$, where $\omega(t)$ corresponds to the weight function of a system of orthogonal polynomials $(p_n)_{n\in\mathbb{N}}$ and is well defined on the closed and compact interval $[a, b]$. Then there exists a linear combination $\ell_n$ of words such that the $n^{th}$ coefficient in Equation (10) satisfies $\alpha_n = \langle \ell_n, S(\hat{x})\rangle$. Furthermore, the sequence $(\ell_n)_{n\in\mathbb{N}}$ satisfies the following recurrence relation*

$$\ell_n = A_n\frac{(p_{n-1}, p_{n-1})}{(p_n, p_n)}e_1 \succ \ell_{n-1} + (A_n a + B_n)\frac{(p_{n-1}, p_{n-1})}{(p_n, p_n)}\ell_{n-1} + C_n\frac{(p_{n-2}, p_{n-2})}{(p_n, p_n)}\ell_{n-2},$$

*with*

$$\ell_0 = \frac{A_0}{(p_0, p_0)}e_{21} \quad and \quad \ell_1 = \frac{A_1}{(p_1, p_1)}(e_{121} + e_{211}) + \frac{A_1 a + B_1}{(p_1, p_1)}e_{21}.$$

**Remark.** *The results in Theorem 3.2 require signatures of $\hat{x} = (t, w(t)x(t))$. However, sometimes one may only have signatures of $\tilde{x} = (t, x(t))$. In Appendix C.2 we propose an alternative method by approximating the weight function as a Taylor series.*

### 3.3 INVERSION TIME COMPLEXITY

The inversion formulae all boil down to evaluating specific linear combinations of signature terms. Evaluating a linear functional has a time complexity linear in the size of the signature. Since this evaluation is repeated for each of the $n$ recovered basis coefficients, the total number of operations is $nm$, where $n$ in the number of basis coefficients and $m$ is the length of the signature truncated at level $n + 2$. For a $d$-dimensional path, length of a step-$N$ signature is $\frac{d^{N+1}-1}{d-1}$, giving the inversion a time complexity of $O(nd^{n+2})$.

## 4 RELATED WORK

**Multivariate time series generation**    Generating multivariate time series has been an active area of research in the past several years, predominantly relying on generative adversarial networks (GANs) (Goodfellow et al., 2014). Simple recurrent neural networks acting as generators and discriminators (Mogren, 2016; Esteban et al., 2017) later evolved into encoder-decoder architectures where the adversarial generation occurs in a learned latent space (Yoon et al., 2019; Pei et al., 2021; Jeon et al., 2022). To generate time series in continuous time, architectures based on neural differential equations in the latent space (Rubanova et al., 2019; Yildiz et al., 2019) have emerged as generalisations of RNNs. More flexible alternatives have since been proposed in the forms of neural controlled differential equations (Kidger et al., 2020) and state space ODEs (Zhou et al., 2023).

**Diffusion models for time series generation**    There are a number of denoising probabilistic diffusion models (DDPMs) currently at the forefront of time series synthesis, such as DiffTime (Ho et al., 2020), which reformulates the constrained time series generation problem in terms of conditional denoising diffusion (Tashiro et al., 2021). Most recently, Diffusion-TS (Yuan & Qiao, 2024) has demonstrated superior performance on benchmark datasets and long time series by disentangling temporal features via a Fourier-based training objective. To learn long-range dependencies, both the aforementioned methods use transformer-based (Vaswani et al., 2017) diffusion functions. Many recent efforts attempt to generalise score-based diffusion to infinite-dimensional function spaces (Kerrigan et al., 2022; Dutordoir et al., 2023; Phillips et al., 2022; Lim et al., 2023). However, unlike their discrete-time counterparts, they have not yet been benchmarked on a variety of real-world temporal datasets. An exception to this is a diffusion framework proposed by Biloš et al. (2023), which synthesises continuous time series by replacing the time-independent noise corruption with samples from a Gaussian process, forcing the diffusion to remain in the space of continuous functions. Another promising approach for training diffusion models in function space is the Denoising Diffusion Operators (DDOs) method (Lim et al., 2023). While it has not been previously applied to time series, our evaluation in Section 5 demonstrates its strong performance in this context. Additionally, there is a growing body of recent literature focusing on application-specific time series generation via diffusion models, such as speech enhancement (Lay et al., 2023; Lemercier et al., 2023), soft sensing (Dai et al., 2023), and battery charging behaviour (Li et al., 2024).

**Signature inversion**    The uniqueness property of signatures mentioned in Section 2.2 has motivated several previous attempts to answer the question of inverting the signature transform, mostly as theoretical contributions focusing on one specific class of paths (Lyons & Xu, 2017; Chang et al., 2016; Lyons & Xu, 2018). The only fast and scalable signature inversion strategy to date is the Insertion method (Chang & Lyons, 2019), which provides an algorithm and theoretical error bounds for inverting piecewise linear paths. It was recently optimised (Fermanian et al., 2023a) and released as a part of the Signatory (Kidger et al., 2019) package. There are also examples of inversion via deep learning (Kidger et al., 2019) and evolutionary algorithms (Buehler et al., 2020), but they provide no convergence guarantees and become largely inefficient when deployed on real-world time series.

## 5 EXPERIMENTS

In Section 5.1, we demonstrate that the newly proposed signature inversion method achieves more accurate reconstructions than the previous Insertion (Chang & Lyons, 2019; Fermanian et al., 2023a) and Optimisation (Kidger et al., 2019) methods. We also analyse the inversion quality and time complexity across different families of orthogonal polynomials. In Section 5.2, we show that (log)signatures,

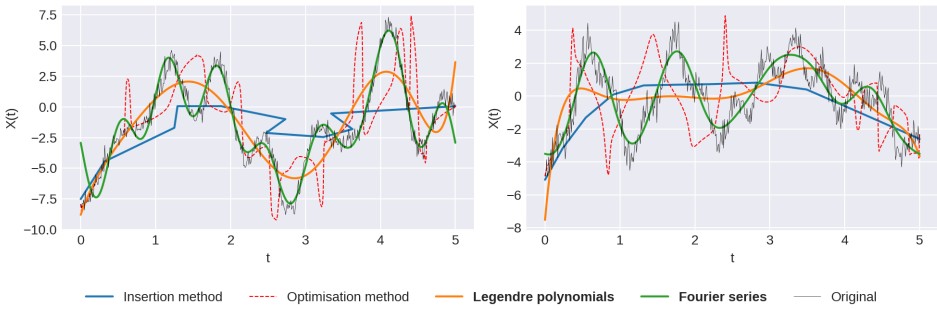

Figure 2: Comparison of different inversion methods.

combined with our closed-form inversion, provide an exceptionally effective embedding for time series diffusion models. Section 5.2.1 visualises the trade-off between the precision of time series representation and model complexity introduced by the choice of the signature truncation level. In Section 5.2.2, we present experiments demonstrating that generating step-4 log-signatures via the pipeline outlined in Figure 1 outperforms other recent time series diffusion models across several standard metrics.

## 5.1 INVERSION EVALUATION

We perform experiments to evaluate the proposed analytical signature inversion formulae derived in Section 3 via several families of orthogonal bases. Using example paths given by sums of random sine waves with injected Gaussian noise, we reconstruct the original paths from their step-12 signatures. Figure 2 compares inversion of these paths via Legendre and Fourier coefficients to the Insertion method (Chang & Lyons, 2019; Fermanian et al., 2023a) and Optimisation method (Kidger et al., 2019), showcasing the improvement in inversion quality provided by our explicit inversion formulae. Figure 3 presents the time consumption against the $L_2$ error as the degree of polynomials increases. Notably, the factor holding the most influence over the reconstruction quality is the truncation level of

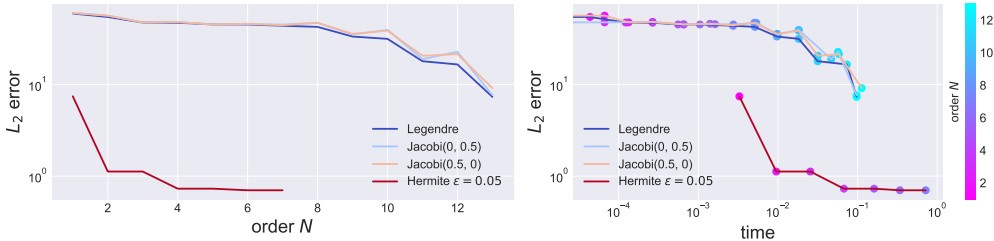

Figure 3: $L_2$ error of signature inversion via orthogonal polynomials with respect to the polynomial order $N$ and time. Error and time are calculated by an average over 15 paths with 200 sample points.

the signature, as it bounds the order of polynomials we can retrieve. We refer the interested reader to a discussion about inversion quality in Appendix D. Namely, Figures 11 and 12 show more examples of inverted signatures using different types of paths and polynomial bases.

## 5.2 GENERATING LONG TIME SERIES

In this section, we introduce the `SigDiffusions` pipeline for generating multivariate time series by the following strategy, also illustrated in Figure 1:

1. Choose an orthogonal basis and order $N$ sufficient to represent the time series with enough detail to retain its meaningful components while smoothing out unnecessary noise.
2. Compute the log-signatures of the time series, augmented as described in Section 3 and truncated at level $N + 2$.

3. Train and sample a score-based diffusion model to generate the log-signatures.

4. Invert the synthetic log-signatures back to time series using our formulae from Section 3.

### 5.2.1 TIME SERIES REPRESENTATION AND MODEL CAPACITY

Here, we highlight the trade-off between model capacity and the faithfulness of the time series representation as given by its truncated Fourier series. Since the signature inversion formulae are exact, the inversion quality depends only on how well the underlying signal is approximated by the retrieved Fourier basis coefficients. When the primary goal is to model the overall shape of a multivariate time series, it is sufficient to truncate the signature, and thus the Fourier series, at a low level. Low truncation levels capture most of this information, smooth out high-order noise, and simplify the generation task. In contrast, modelling signals with high-frequency components requires higher signature truncation levels, raising the complexity of the diffusion task. As the signature size grows exponentially with the truncation level, generating highly oscillatory time series with high fidelity becomes increasingly constrained by the model's capacity. To illustrate this, we show the generated samples from `SigDiffusions` trained on a noisy Lotka–Volterra system with different log-signature truncation levels in Figure 4. At each truncation level, the log-signatures become progressively more difficult to generate accurately.

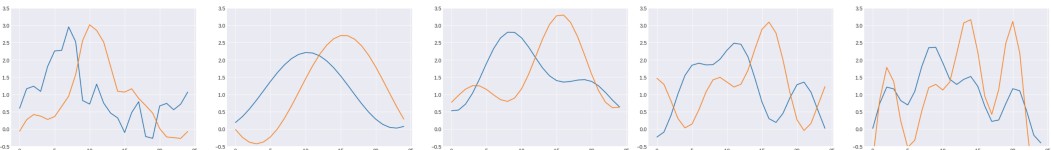

Figure 4: Time series representation and model capacity trade-off. Left to right: real sample from a noisy Lotka–Volterra system, sample generated by `SigDiffusion` with signature truncation level 3, 4, 5, 6.

### 5.2.2 GENERATING STEP-4 LOG-SIGNATURES

We now demonstrate the exceptional ability of signatures to capture the shape and cross-channel dependencies of time series at low truncation levels. We show that `SigDiffusions` applied to step-4 log-signatures, combined with Fourier inversion, outperform state-of-the-art diffusion-based models on the task of generating 1000-point-long time series.

**Datasets**   We perform experiments on five different time series datasets: **Sines** (Yoon et al., 2019) - a dataset of 5-dimensional sine curves with randomly sampled frequency and phase, **Predator-prey** (Biloš et al., 2023) - a two-dimensional continuous system evolving according to a set of ODEs, **Household Electric Power Consumption (HEPC)** (UCI Machine Learning Repository, 2024) - the univariate *voltage* feature from a real-world dataset of household power consumption, **Exchange Rates** (Lai et al., 2018; Lai, 2017) - a real-world dataset containing daily exchange rates of 8 currencies, and **Weather** (Kolle, 2024) - a real-world dataset reporting weather measurements.

**Metrics**   We use the metrics established in Yoon et al. (2019). The **Discriminative Score** reports the out-of-sample accuracy of a recurrent neural network (RNN) classifier trained to distinguish between real and generated time series. To improve readability, we report values offset by $0.5$, so that in the ideal case, where real and generated samples are indistinguishable by the classifier, this metric will approach $0$. The **Predictive Score** measures the loss of a next-point predictor RNN trained exclusively on synthetic data, with the loss evaluated on the real test data set. In the case of univariate time series, the RNN predicts the value of the time series 20 time points ahead, instead of predicting the value of an unseen channel. We also run the Kolmogorov-Smirnov (KS) test on marginal distributions of random batches of ground truth and generated paths. We repeat this test 1000 times with a batch size of 64 and report the mean KS score with the mean Type I error for a 5% significance threshold. Since the cross-channel terms of the log-signature are not necessary for the inversion methods, we generate a concatenated vector of the log-signatures of each separate dimension plus the augmentation described in Section 3.

Table 1: Results for generating time series of length 1000.

| Dataset | Model | Discriminative Score | Predictive Score |
|---|---|---|---|
| Sines | SigDiffusion (ours) | 0.100±.026 | 0.191±.004 |
| | DDO ($\gamma = 1$) | **0.040±.025** | **0.187±.003** |
| | Diffusion-TS | 0.291±.070 | 0.365±.006 |
| | CSPD-GP (RNN) | 0.468±.030 | 0.244±.012 |
| | CSPD-GP (Transformer) | 0.493±.003 | 0.390±.023 |
| Predator-prey | SigDiffusion (ours) | 0.184±.058 | **0.056±.000** |
| | DDO ($\gamma = 10$) | 0.211±.080 | **0.056±.000** |
| | Diffusion-TS | 0.500±.000 | 0.482±.023 |
| | CSPD-GP (RNN) | **0.168±.024** | 0.057±.000 |
| | CSPD-GP (Transformer) | 0.498±.003 | 0.919±.002 |
| HEPC | SigDiffusion (ours) | **0.070±.032** | 0.050±.012 |
| | DDO ($\gamma = 1$) | 0.081±.019 | **0.044±.001** |
| | Diffusion-TS | 0.438±.057 | 0.066±.022 |
| | CSPD-GP (RNN) | 0.415±.045 | 0.108±.002 |
| | CSPD-GP (Transformer) | 0.500±.000 | 0.551±.028 |
| Exchange Rates | SigDiffusion (ours) | **0.278±.062** | **0.057±.001** |
| | DDO ($\gamma = 1$) | 0.326±.102 | 0.094±.004 |
| | Diffusion-TS | 0.401±.196 | 0.120±.016 |
| | CSPD-GP (RNN) | 0.500±.001 | 0.273±.100 |
| | CSPD-GP (Transformer) | 0.500±.000 | 0.432±.074 |
| Weather | SigDiffusion (ours) | **0.350±.080** | **0.168±.001** |
| | DDO ($\gamma = 10$) | 0.356±.196 | 0.307±.007 |
| | Diffusion-TS | 0.498±.003 | 0.438±.035 |
| | CSPD-GP (RNN) | 0.500±.000 | 0.505±.007 |
| | CSPD-GP (Transformer) | 0.500±.000 | 0.490±.000 |

**Benchmarks**  Table 1 lists the time series generation performance metrics compared with four recent diffusion model architectures specifically designed to handle long or continuous-time paths:

- **Diffusion-TS (Yuan & Qiao, 2024)** This model introduces a novel Fourier-based training objective to disentangle temporal features of different seasonalities. This interpretable decomposition strategy makes the model particularly robust to varying time series lengths, demonstrating strong performance relative to other benchmarks as the training time series become longer.

- **CSPD-GP (Biloš et al., 2023)** This approach replaces the time-independent noise corruption mechanism with samples drawn from a Gaussian process, effectively modelling diffusion on time series as a process occurring within the space of continuous functions. CSPD-GP (RNN) and CSPD-GP (Transformer) refer to score-based diffusion models with the score function either being an RNN or a transformer.

- **Denoising Diffusion Operators (DDO) (Lim et al., 2023)** These models generalise diffusion models to function spaces with a Hilbert space-valued Gaussian process to perturb the input data. Additionally, they use neural operators for the score function, ensuring consistency with the underlying function space formulation. DDO's kernel smoothness hyperparameter $\gamma$ is tuned and reported for each dataset.

We compute the metrics using 1000 samples from each model. Table 2 shows the model sizes and training times, demonstrating that `SigDiffusion` outperforms the other models while also having the most efficient architecture. Table 4 in the Appendix evaluates the time series marginals using the KS test. More details about the experimental setup can be found in Appendix E.

Table 2: Comparison of model sizes.

| Dataset | Model | Parameters | Training Time | Sampling Time |
|---------|-------|-----------|---------------|---------------|
| Sines | SigDiffusion (ours) | 229K | 8 min | 11 sec |
| | DDO ($\gamma = 1$) | 4.12M | 3.6 h | 42 min |
| | Diffusion-TS | 4.18M | 57 min | 15 min |
| | CSPD-GP (RNN) | 759K | 9 min | 1 min |
| | CSPD-GP (Transformer) | 973K | 15 min | 5 min |
| Predator-prey | SigDiffusion (ours) | 211K | 8 min | 12 sec |
| | DDO ($\gamma = 10$) | 4.12M | 3.5 h | 42 min |
| | Diffusion-TS | 4.17M | 55 min | 14 min |
| | CSPD-GP (RNN) | 758K | 8 min | 1 min |
| | CSPD-GP (Transformer) | 972K | 16 min | 5 min |
| HEPC | SigDiffusion (ours) | 206K | 8 min | 12 sec |
| | DDO ($\gamma = 1$) | 4.12M | 2.6 h | 42 min |
| | Diffusion-TS | 4.17M | 50 min | 15 min |
| | CSPD-GP (RNN) | 758K | 4 min | 1 min |
| | CSPD-GP (Transformer) | 972K | 9 min | 5 min |
| Exchange rates | SigDiffusion (ours) | 247 K | 9 min | 11 sec |
| | DDO ($\gamma = 1$) | 4.12M | 3.6 h | 42 min |
| | Diffusion-TS | 4.29 M | 1.2 h | 20 min |
| | CSPD-GP (RNN) | 760 K | 11 min | 1 min |
| | CSPD-GP (Transformer) | 974 K | 15 min | 6 min |
| Weather | SigDiffusion (ours) | 282K | 8 min | 12 sec |
| | DDO ($\gamma = 10$) | 4.12M | 3.6 h | 42 min |
| | Diffusion-TS | 4.3 M | 1.4 h | 20 min |
| | CSPD-GP (RNN) | 763 K | 14 min | 1 min |
| | CSPD-GP (Transformer) | 975 K | 17 min | 6 min |

# 6 CONCLUSION AND FUTURE WORK

In this paper, we introduced `SigDiffusion`, a new diffusion model that gradually perturbs and denoises log-signature embeddings of long time series, preserving their Lie algebraic structure. To recover the path from its log-signature, we proved that the coefficients of the expansion of a path in a given basis, such as Fourier or orthogonal polynomials, can be expressed as explicit linear functionals on the signature, or equivalently as polynomial functions on the log-signature. These results provide explicit signature inversion formulae, representing a major improvement over signature inversion algorithms previously proposed in the literature. Finally, we demonstrated how combining `SigDiffusion` with these inversion formulae provides a powerful generative approach for time series that is competitive with state-of-the-art diffusion models for temporal data.

As this is the first work on diffusion models for time series using signature embeddings, there are still many research directions to explore. To mitigate the rapid growth in the number of required signature features for high-frequency signals described in Section 5.2.1, future work could explore alternative embeddings to the signature. For example, other types of path developments derived from rough path theory, which embed temporal signals into (compact) Lie groups, such as those proposed by Cass & Turner (2024), may offer a more parsimonious representation. These alternatives retain many of the desirable properties of signatures, including the availability of a flat-space Lie algebra. However, it remains unclear how an inversion mechanism would work in these cases. Finally, it would be interesting to understand how *discrete-time signatures* (Diehl et al., 2023) could be leveraged to encode *discrete sequences* on Lie groups and potentially perform diffusion-based generative modelling for text.

ACKNOWLEDGMENTS

Barbora Barancikova is supported by UK Research and Innovation [UKRI Centre for Doctoral Training in AI for Healthcare grant number EP/S023283/1].

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

APPENDIX

This appendix is structured in the following way. In Appendix A we complement the material presented in Section 2 with additional details on the signature. In Appendix B, we provide examples of orthogonal polynomial families one can use for signature inversion due to the derived inversion formulae in Section 3. In Appendix C we provide proofs for the signature inversion Theorem 3.1 and Theorem 3.2. Appendix D contains additional examples and discussion about the quality of signature inversion by different bases. Appendix E provides details on the implementation of experiments.

# A  ADDITIONAL DETAILS ON THE SIGNATURE

In this section, we establish the foundational algebraic framework for signatures in Appendix A.1. We then provide a mathematically rigorous definition of the (log)signature in Appendix A.2, building upon the introduction in Section 2. The section concludes with illustrative signature computation examples in Appendix A.3.

## A.1  ALGEBRAIC SETUP

For any positive integer $n \in \mathbb{N}$, we consider the *truncated tensor algebra* over $\mathbb{R}^d$

$$T^n(\mathbb{R}^d) := \bigoplus_{k=0}^{n} (\mathbb{R}^d)^{\otimes k},$$

where $\otimes$ denotes the outer product of vector spaces. For any scalar $\alpha \in \mathbb{R}$, we denote by $T_\alpha^n(\mathbb{R}^d) = \{A \in T^n(\mathbb{R}^d) : A_0 = \alpha\}$ the hyperplane of elements in $T^n(\mathbb{R}^d)$ with the $0^{th}$ term equal to $\alpha$.

$T^n(\mathbb{R}^d)$ is a non-commutative algebra when endowed with the tensor product $\cdot$ defined for any two elements $A = (A_0, A_1, ..., A_n)$ and $B = (B_0, B_1, ...B_n)$ of $T^n(\mathbb{R}^d)$ as follows

$$A \cdot B = (C_0, C_1, ..., C_n) \in T^n(\mathbb{R}^d), \quad \text{where} \quad C_k = \sum_{i=0}^{k} A_i \otimes B_{k-i} \in (\mathbb{R}^d)^{\otimes k}. \quad (11)$$

The standard basis of $\mathbb{R}^d$ is denoted by $e_1, e_2, ..., e_d$. We will refer to these basis elements as *letters*. Elements of the induced standard basis of $T^n(\mathbb{R}^d)$ are often referred to as *words* and abbreviated

$$e_{i_1 i_2 ... i_k} = e_{i_1} \otimes e_{i_2} \otimes ... \otimes e_{i_k}, \quad \text{for} \quad 1 \leq i_1, ..., i_k \leq d \text{ and } 0 \leq k \leq n.$$

We will make use of the dual pairing notation $\langle e_{i_1 i_2 ... i_k}, A \rangle \in \mathbb{R}$ to denote the $(i_1, ..., i_k)^{th}$ element of a tensor $A \in T^n(\mathbb{R}^d)$. This pairing is extended by linearity to any linear combination of words.

Following Reutenauer (2003), the truncated tensor algebra $T^n(\mathbb{R}^d)$ carries several additional algebraic structures.

Firstly, it is a *Lie algebra*, where the Lie bracket is the commutator

$$[A, B] = A \cdot B - B \cdot A \quad \text{for } A, B \in T^n(\mathbb{R}^d).$$

We denote by $\mathcal{L}^n(\mathbb{R}^d)$ the smallest Lie subalgebra of $T^n(\mathbb{R}^d)$ containing $\mathbb{R}^d$. We note that the Lie algebra $\mathcal{L}^n(\mathbb{R}^d)$ is a vector space of dimension $\beta(d, n)$ with

$$\beta(d, n) = \sum_{k=1}^{n} \frac{1}{k} \sum_{i|k} \mu \left( \frac{k}{i} \right) d^i,$$

where $\mu$ is the Möbius function (Reutenauer, 2003). Bases of this space are known as *Hall bases* (Reutenauer, 2003; Reizenstein, 2017). One of the most well-known bases is the *Lyndon basis* indexed by *Lyndon words*. A Lyndon word is a word occurring lexicographically earlier than any word obtained by cyclically rotating its elements.

Secondly, $T^n(\mathbb{R}^d)$ is also a commutative algebra with respect to the *shuffle product* ⧢. On basis elements, the shuffle product of two words of length $r$ and $s$ (with $r + s \leq n$) is the sum over the

$\binom{r+s}{s}$ ways of interleaving the two words. For a more formal definition, see Reutenauer (2003, Section 1.4).

Related to the shuffle product is the *right half-shuffle product* $\succ$ defined recursively as follows: for any two words $e_{i_1 \dots i_r}$ and $e_{j_1 \dots j_s}$ and letter $e_j$

$$e_{i_1 \dots i_r} \succ e_j = e_{i_1 \dots i_r j} \quad \text{and} \quad e_{i_1 \dots i_r} \succ e_{j_1 \dots j_s} = (e_{i_1 \dots i_r} \succ e_{j_1 \dots j_{s-1}} + e_{j_1 \dots j_{s-1}} \succ e_{i_1 \dots i_r}) \cdot e_{j_s}.$$

The right half-shuffle product will be useful for carrying out computations in the next section. Note that the following relation between shuffle and right half-shuffle products holds (Salvi et al., 2023)

$$e_{i_1 \dots i_r} \shuffle e_{j_1 \dots j_s} = e_{i_1 \dots i_r} \succ e_{j_1 \dots j_s} + e_{j_1 \dots j_s} \succ e_{i_1 \dots i_r}.$$

Equipped with this algebraic setup, we can now introduce the signature.

## A.2 THE (LOG)SIGNATURE

Let $x : [0, 1] \to \mathbb{R}^d$ be a smooth path. The *step-$n$ signature* $S^{\leq n}(x)$ of $x$ is defined as the following collection of iterated integrals

$$S^{\leq n}(x) = (1, S_1(x), \dots, S_n(x)) \in T_1^n(\mathbb{R}^d) \tag{12}$$

where

$$S_k(x) = \int_{0 \leq t_1 < \dots < t_k \leq 1} dx_{t_1} \otimes \dots \otimes dx_{t_k} \in (\mathbb{R}^d)^{\otimes k} \quad \text{for } 1 \leq k \leq n.$$

An important property of the signature is usually referred to as the *shuffle identity*. This result is originally due to Ree (1958). For a modern proof see (Cass & Salvi, 2024, Theorem 1.3.10).

**Lemma A.0.1** (Shuffle identity). *(Ree, 1958) Let $x : [0, 1] \to \mathbb{R}^d$ be a smooth path. For any two words $e_{i_1 \dots i_r}$ and $e_{j_1 \dots j_s}$, with $0 \leq r, s \leq n$, the following two identities hold*

$$\left\langle e_{i_1 \dots i_r} \shuffle e_{j_1 \dots j_s}, S^{\leq n}(x) \right\rangle = \left\langle e_{i_1 \dots i_r}, S^{\leq n}(x) \right\rangle \left\langle e_{j_1 \dots j_s}, S^{\leq n}(x) \right\rangle,$$

$$\left\langle e_{i_1 \dots i_r} \succ e_{j_1 \dots j_s}, S^{\leq n}(x) \right\rangle = \int_0^1 \left\langle e_{i_1 \dots i_r}, S^{\leq n}(x)_t \right\rangle d \left\langle e_{j_1 \dots j_s}, S^{\leq n}(x)_t \right\rangle,$$

*where $S^{\leq n}(x)_t$ is the step-$n$ signature of the path $x$ restricted to the interval $[0, t]$.*

An example of simple computations using the shuffle identity is presented in Appendix A.3.

Moreover, it turns out that the signature is more than just a generic element of $T_1^n(\mathbb{R}^d)$; in fact, its range has the structure of a Lie group as we shall explain next. Recall that the tensor exponential $\exp$ and the tensor logarithm $\log$ are maps from $T^n(\mathbb{R}^d)$ to itself defined as follows

$$\exp(A) := \sum_{k \geq 0} \frac{1}{k!} A^{\otimes k} \quad \text{and} \quad \log(\mathbf{1} + A) = \sum_{k \geq 1} \frac{(-1)^{k-1}}{k} (A)^{\otimes k} \tag{13}$$

where $\mathbf{1} = (1, 0, \dots, 0) \in T^n(\mathbb{R}^d)$. It is a well-known fact that $\exp : T_0^n(\mathbb{R}^d) \to T_1^n(\mathbb{R}^d)$ and $\log : T_1^n(\mathbb{R}^d) \to T_0^n(\mathbb{R}^d)$ are mutually inverse.

The *step-$n$ free nilpotent Lie group* is the image of the free Lie algebra under the exponential map

$$\mathcal{G}^n(\mathbb{R}^d) = \exp(\mathcal{L}^n(\mathbb{R}^d)) \subset T_1^n(\mathbb{R}^d). \tag{14}$$

As its name suggests, $\mathcal{G}^n(\mathbb{R}^d)$ is a Lie group and plays a central role in the theory of rough paths (Friz & Victoir, 2010).

Here comes the connection with signatures. It is established by the following fundamental result due to Chen (1957; 1958), which can also be viewed as a consequence of Chow's results in Chow (1939).

**Lemma A.0.2** (Chen–Chow). *(Chen, 1957; 1958; Chow, 1939) The step-$n$ free nilpotent Lie group $\mathcal{G}^n(\mathbb{R}^d)$ is precisely the image of the step-$n$ signature map in Equation (12) when the latter is applied to all smooth paths in $\mathbb{R}^d$*

$$\mathcal{G}^n(\mathbb{R}^d) = \{S^{\leq n}(x) \mid x : [0, 1] \to \mathbb{R}^d \text{ smooth}\}.$$

## A.3 Simple examples of signature computations

In the following examples, we alter the notation so that for a path $x : [a, t] \to \mathbb{R}^d$, the tensor representing the $k$-th level of the signature computed on an interval $[a, t]$ is denoted as

$$S(x)_{a,t}^{(k)} = (S(x)_{a,t}^{i_1,\dots,i_k} : i_1, \dots, i_k \in \{1, \dots, d\}) \in (\mathbb{R}^d)^{\otimes k}. \tag{15}$$

Furthermore, we can express the value of $S(x)_{a,t}^{(k)}$ at a particular set of indices $i_1, \dots, i_k \in \{1, \dots d\}$ as a *k-fold iterated integral*

$$S(x)_{a,t}^{i_1,\dots,i_k} = \int_{a < t_1 < \cdots < t_k < t} dx_{t_1}^{i_1} \dots dx_{t_k}^{i_k}. \tag{16}$$

We assume that the signature is always truncated at a sufficiently high level $n$, allowing us to denote the step-$n$ signature simply as

$$S(x)_{a,t} = (1, S(x)_{a,t}^{(1)}, S(x)_{a,t}^{(2)}, S(x)_{a,t}^{(3)}, \dots, S(x)_{a,t}^{(n)}) \in T_1^n(\mathbb{R}^d). \tag{17}$$

**Example A.1** (Geometric interpretation of a 2-dimensional path)**.** *Consider a path* $\hat{x} : [0, 9] \to \mathbb{R}^2$, *where* $\hat{x} = (x_t^1, x_t^2) = (t, x(t))$. *Here,* $x(t)$ *is defined as*

$$x_t^2 = x(t) = \begin{cases} \sqrt{3}t & t \in [0, 2] \\ 2\sqrt{3} & t \in [2, 8] \\ \sqrt{3}t - 6\sqrt{3} & t \in [8, 9] \end{cases},$$

*which is continuous and piecewise differentiable. In this case,* $\dot{x}_t^1 = 1$, *and* $\dot{x}_t^2$ *can be expressed as*

$$\dot{x}_t^2 = \dot{x}(t) = \begin{cases} \sqrt{3} & t \in (0, 2) \\ 0 & t \in (2, 8) \\ \sqrt{3} & t \in (8, 9) \end{cases}.$$

*One can compute the step-$n$ signature of* $\hat{x}$ *as*

$$S(\hat{x})_{0,9} = (1, S(\hat{x})_{0,9}^{(1)}, S(\hat{x})_{0,9}^{(2)}, S(\hat{x})_{0,9}^{(3)}, \dots, S(\hat{x})_{0,9}^{(n)})$$
$$= (1, S(\hat{x})_{0,9}^1, S(\hat{x})_{0,9}^2, S(\hat{x})_{0,9}^{1,2}, S(\hat{x})_{0,9}^{2,1}, S(\hat{x})_{0,9}^{1,1,1}, \dots, S(\hat{x})_{0,9}^{i_1,\dots,i_n}),$$

*where*

$$S(\hat{x})_{0,9}^1 = \int_{0<s<9} dx_s^1 = x_9^1 - x_0^1 = 9$$

$$S(\hat{x})_{0,9}^2 = \int_{0<s<9} dx_s^2 = x_9^2 - x_0^2 = 3\sqrt{3}$$

$$S(\hat{x})_{0,9}^{1,1} = \int_{0<r<s<9} dx_r^1 dx_s^1 = \int_{0<s<9} x_s^1 dx_s^1 = \frac{1}{2}\left(x_s^1\right)^2 \Big|_0^9 = \frac{81}{2}$$

$$S(\hat{x})_{0,9}^{1,2} = \int_{0<r<s<9} dx_r^1 dx_s^2 = \int_{0<s<9} s dx_s^2 = \int_{0<s<9} s\dot{x}_s^2 ds = \frac{\sqrt{3}}{2}s^2 \Big|_0^2 + \frac{\sqrt{3}}{2}s^2 \Big|_8^9 = \frac{21}{2}\sqrt{3}$$

$$S(\hat{x})_{0,9}^{2,1} = \int_{0<r<s<9} dx_r^2 dx_s^1 = \int_{0<s<9} x_s^2 ds = \frac{\sqrt{3}}{2}s^2 \Big|_0^2 + 2\sqrt{3}s \Big|_2^8 + \frac{\sqrt{3}}{2}s^2 - 6\sqrt{3}s \Big|_8^9 = \frac{33}{2}\sqrt{3}$$

$$S(\hat{x})_{0,9}^{2,2} = \int_{0<r<s<9} dx_r^2 dx_s^2 = \int_{0<s<9} x_s^2 dx_s^2 = \frac{1}{2}\left(x_s^2\right)^2 \Big|_0^9 = \frac{27}{2}.$$

*From Figure 5, let* $A_-$ *and* $A_+$ *represent the signed value of the shaded region. The signed Lévy area of the path is defined as* $A_- + A_+$. *In this case, the signed Lévy area is* $-3\sqrt{3}$. *Surprisingly,*

$$\frac{1}{2}\left(S(\hat{x})_{0,9}^{1,2} - S(\hat{x})_{0,9}^{2,1}\right) = \frac{1}{2}\left(\frac{21}{2}\sqrt{3} - \frac{33}{2}\sqrt{3}\right) = -3\sqrt{3} = A_- + A_+,$$

*which is exactly the signed Lévy area.*

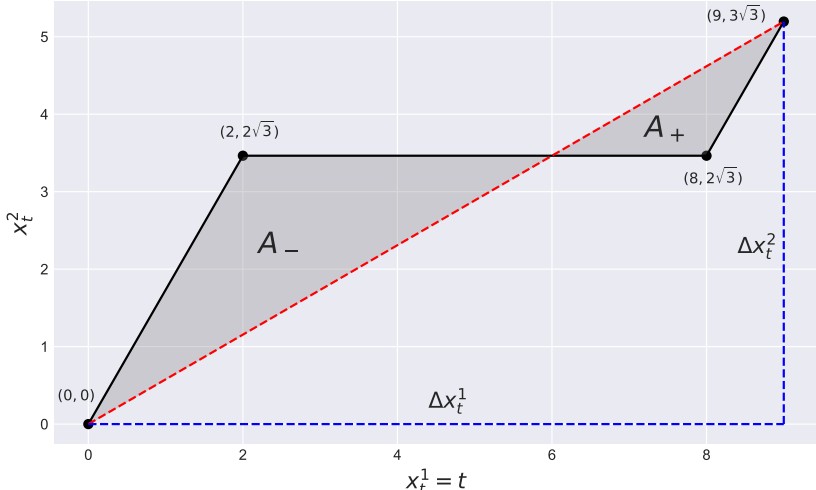

Figure 5: Path in Example A.1. The shaded region represents the signed Lévy area.

Another important example is given by the signature of linear paths.

**Example A.2** (Signatures of linear paths). *Suppose there is a linear path $x : [a, b] \to \mathbb{R}^d$. Then the path $x$ is linear in terms of $t$, i.e.*

$$x_t = x_a + \frac{t - a}{b - a} \left( x_b - x_a \right).$$

*It follows that its derivative can be written as*

$$dx_t = \frac{(x_b - x_a)}{b - a} dt.$$

*Recalling the definition of a signature, it holds that*

$$
\begin{aligned}
S(x)_{a,b}^{i_1,\dots,i_k} &= \int_{a < t_1 < \cdots < t_k < b} dx_{t_1}^{i_1} \dots dx_{t_k}^{i_k} \\
&= \frac{\prod_{j=1}^{k} \left( x_b^{i_j} - x_a^{i_j} \right)}{(b - a)^k} \int_{a < t_1 < \cdots < t_k < b} dt_1 \dots dt_k \\
&= \frac{\prod_{j=1}^{k} \left( x_b^{i_j} - x_a^{i_j} \right)}{(b - a)^k} \frac{(b - a)^k}{k!} \\
&= \frac{\prod_{j=1}^{k} \left( x_b^{i_j} - x_a^{i_j} \right)}{k!}.
\end{aligned}
$$

*Therefore, the whole step-$n$ signature can be expressed as a tensor exponential of the linear increment $x_b - x_a$*

$$
S(x)_{a,b}^{(k)} = \frac{(x_b - x_a)^{\otimes k}}{k!},
$$

$$
S(x)_{a,b} = \sum_{k=0}^{n} \frac{(x_b - x_a)^{\otimes k}}{k!}
$$

$$
= \exp_\otimes \left( x_b - x_a \right).
$$

Chen's identity in Lemma 2.0.1 is one of the most fundamental algebraic properties of the signature as it describes the behaviour of the signature under the concatenation of paths.

**Definition A.1** (Concatenation). *Consider two smooth paths $x : [a, b] \to \mathbb{R}^d$ and $y : [b, c] \to \mathbb{R}^d$. Define the concatenation of $x$ and $y$, denoted by $x * y$ as a path $[a, c] \to \mathbb{R}^d$*

$$(x * y)_t := \begin{cases} x_t & \text{if } a \leq t \leq b \\ x_b - y_b + y_t & \text{if } b \leq t \leq c \end{cases}.$$

Chen's identity in Lemma 2.0.1 provides a method to simplify the analysis of longer paths by converting them into manageable shorter ones. If we have a smooth path $x : [t_0, t_n] \to \mathbb{R}^d$, then inductively, we can decompose the signature of $x$ to

$$S(x)_{t_0, t_n} = S(x)_{t_0, t_1} \cdot S(x)_{t_1, t_2} \cdot \cdots \cdot S(x)_{t_{n-1}, t_n}.$$

Moreover, if we have a time series $(t_0, x_0), ..., (t_n, x_n) \in \mathbb{R}^{d+1}$, we can treat $x$ as a piecewise linear path interpolating the data. Based on Example A.2, one can observe that

$$S(x)_{t_0, t_n} = \exp_\otimes (x_{t_1} - x_{t_0}) \cdot \exp_\otimes (x_{t_2} - x_{t_1}) \cdot \cdots \cdot \exp_\otimes (x_{t_n} - x_{t_{n-1}}),$$

which is widely used in Python packages such as `esig` or `iisignature`.

**Example A.3** (Example of shuffle identity). *Consider a smooth path $x = (x_t^1, x_t^2) : [a, b] \to \mathbb{R}^2$. By integration by parts, we have*

$$\begin{aligned} \langle e_1, S(x)_{a,b} \rangle \langle e_2, S(x)_{a,b} \rangle &= \int_{a<t<b} dx_t^1 \int_{a<t<b} dx_t^2 \\ &= \int_{a<t<b} \dot{x}_t^1 dt \int_{a<t<b} \dot{x}_t^2 dt \\ &\overset{\text{by parts}}{=} \int_{a<t<b} \langle e_2, S(x)_{a,t} \rangle \dot{x}_t^1 dt + \int_{a<t<b} \langle e_1, S(x)_{a,t} \rangle \dot{x}_t^2 dt \\ &= \langle e_{2,1}, S(x)_{a,b} \rangle + \langle e_{1,2}, S(x)_{a,b} \rangle. \end{aligned}$$

*By the shuffle identity, we have*

$$\begin{aligned} \langle e_1, S(x)_{a,b} \rangle \langle e_2, S(x)_{a,b} \rangle &= \langle e_1, S(x)_{a,b} \rangle \langle e_2, S(x)_{a,b} \rangle \\ &= \langle e_1 \shuffle e_2, S(x)_{a,b} \rangle \\ &= \langle e_{1,2} + e_{2,1}, S(x)_{a,b} \rangle \\ &= \langle e_{1,2}, S(x)_{a,b} \rangle + \langle e_{2,1}, S(x)_{a,b} \rangle, \end{aligned}$$

*which is exactly the same as what we derived via integration by parts.*

**Example A.4** (Example of half-shuffle computations). *Consider a two-dimensional, real-valued smooth path $\hat{x} = (t, x(t)) : [a, b] \to \mathbb{R}^2$ with $x(a) = 0$. The elements of the first truncation level of the signature can be retrieved as follows:*

$$\langle e_1, S(\hat{x})_{a,t} \rangle = \int_a^t ds = t - a, \quad \langle e_2, S(\hat{x})_{a,t} \rangle = \int_a^t d(x(s)) = x(t) - x(a) = x(t).$$

*Then, one can express all integrals in terms of powers of $t - a$ and $x(t)$ by signatures of $\hat{x}$. For example, let $n, m \in \mathbb{N}_0$,*

$$\begin{aligned} \int_a^b (t-a)^n x(t)^m dt &= \int_a^b (t-a)^n x(t)^m d(t-a) \\ &= \int_a^b (\langle e_1, S(\hat{x})_{a,t} \rangle)^n (\langle e_2, S(\hat{x})_{a,t} \rangle)^m d(\langle e_1, S(\hat{x})_{a,t} \rangle) \\ &= \langle (e_1^{\shuffle n} \shuffle e_2^{\shuffle m}) \succ e_1, S(\hat{x})_{a,b} \rangle. \end{aligned}$$

## B  ORTHOGONAL POLYNOMIALS AND FOURIER SERIES

In this section, we introduce the background material on orthogonal polynomials and the Fourier series necessary for the signature inversion formulae presented in the next section.

## B.1 ORTHOGONAL POLYNOMIALS

### B.1.1 INNER PRODUCT AND ORTHOGONALITY

Consider a dot product $(x, y) = \sum_{i=1}^{n} x_i y_i$, where $x, y \in \mathbb{R}^n$. For some weights $w_1, \cdots, w_n \in \mathbb{R}_+$, we can also define the weighted dot product $(x, y)_w = \sum_{i=1}^{n} w_i x_i y_i$, where $(\cdot, \cdot)_w$ can be written as $(\cdot, \cdot)$ for simplicity.

For $p \in [1, \infty)$, $L_w^p(\Omega)$ is the linear space of measurable functions from $\Omega$ to $\mathbb{R}$ such that their weighted $p$-norms are bounded, i.e.

$$L_w^2(\Omega) = \left\{ v \text{ is measurable in } \Omega \,\middle|\, \int_\Omega |v(t)|^2 w(t) dt < \infty \right\}.$$

For example, let $d\alpha$ be a non-negative Borel measure supported on the interval $[a, b]$ and $\mathbb{V} = L_w^2(a, b)$. One can define $(f, g) = \int_a^b f(t) g(t) d\alpha(t)$ as a Stieltjes integral for all $f, g \in \mathbb{V}$. Note that if $\alpha(t)$ is absolutely continuous, which will be the setting throughout this section, then one can find a weight density $w(t)$ such that $d\alpha(t) = w(t) dt$. In this case, the definition of the inner product over a function space reduces to an integral with respect to a weight function, i.e.,

$$(f, g) = \int_a^b f(t) g(t) w(t) dt.$$

We can then consider an orthogonal polynomial system to be orthogonal with respect to the *weight* function $w$. We denote $\mathbb{P}[t] \subset L_w^2(\Omega)$ as the space of all polynomials. A polynomial of degree $n$, $p \in \mathbb{P}_n[t]$, is *monic* if the coefficient of the $n$-th degree is one.

**Definition B.1** (Orthogonal polynomials). *For an arbitrary vector space $\mathbb{V}$, $u$ and $v$ are orthogonal if $(u, v) = 0$ for all $u, v \in \mathbb{V}$. When $\mathbb{V} = \mathbb{P}[t]$, a sequence of polynomials $(p_n)_{n \in \mathbb{N}} \in \mathbb{P}[t]$ is called orthogonal with respect to a weight function $w$ if for all $m \neq n$,*

$$(p_n, p_m) = \int p_n(t) p_m(t) w(t) dt = 0,$$

*where $deg(p_n) = n$ is the degree of a polynomial. Furthermore, we say the sequence of orthogonal polynomials is orthonormal if $(p_n, p_n) = 1$ for all $n \in \mathbb{N}$.*

For simplification, the inner product notation $(\cdot, \cdot)$ will be used without specifying the integral formulation for the orthogonal polynomials. To construct a sequence of orthogonal polynomials in Definition B.1, one can follow the Gram-Schmidt orthogonalisation process, which is stated below.

**Theorem B.1** (Gram-Schmidt orthogonalisation). *The polynomial system $(p_n)_{n \in \mathbb{N}}$ with respect to the inner product $(\cdot, \cdot)$ can be constructed recursively by*

$$p_0 = 1, \qquad p_n = t^n - \sum_{i=1}^{n-1} \frac{(t^n, p_i)}{(p_i, p_i)} p_i \quad \text{for } n \geq 1. \tag{18}$$

From the orthogonalisation process in Theorem B.1, we can see that the $n$-th polynomial $p_n$ has degree $n$ exactly, which means $(p_n)_{n \in \mathbb{N}}$ is a basis spanning $\mathbb{P}[t]$. Furthermore, the orthogonal construction makes the orthogonal polynomial system an orthogonal basis with respect to the corresponding inner product. The following proposition forms an explicit expression for coefficients of $(p_k)_{k \in \{0, \cdots, n\}}$ in an arbitrary $n$-th degree polynomial.

**Proposition B.1.1** (Orthogonal polynomial expansion). *Consider an arbitrary polynomial $x(t) \in \mathbb{P}_n[t]$. One can express $x(t)$ by a sequence of orthogonal polynomials $(p_k)_{k \in \{0, \cdots, n\}}$, i.e.,*

$$x(t) = \sum_{k=0}^{n} \frac{(p_k, x)}{(p_k, p_k)} p_k(t).$$

**Remark.** *We have stated the orthogonal polynomial expansion for $x \in \mathbb{P}_n[t]$. In general, by the closure of orthogonal polynomial systems in $L_w^2(a, b)$, arbitrary $f \in L_w^2(a, b)$ can be written as an infinite sequence of orthogonal polynomials.*

$$f(t) = \sum_{k=0}^{\infty} \frac{(p_k, f)}{(p_k, p_k)} p_k(t).$$

*The $N$-th degree approximation of $f$ is the best approximating polynomial with a degree less or equal to $N$, denoted by*

$$P_N f(t) = \sum_{k=0}^{N} \frac{(p_k, f)}{(p_k, p_k)} p_k(t). \tag{19}$$

### B.1.2 Basic properties

Here, we will list the main properties of orthogonal polynomials significant for our application.

#### THE THREE-TERM RECURRENCE RELATION

**Theorem B.2** (Three-term recurrence relation). *A system of orthogonal polynomials $(p_n)_{n \in \mathbb{N}}$ with respect to a weight function $w$ satisfies the three-term recurrence relation.*

$$p_0(t) = 1, \quad p_1(t) = A_1 t + B_1, \quad p_{n+1}(t) = (A_{n+1} t + B_{n+1}) p_n(t) + C_{n+1} p_{n-1}(t),$$

*for all $n \in \mathbb{N}$, and $A_i > 0$ for all $i \in \mathbb{N}_0$.*

Before proving the recurrence relation, we will first show that an orthogonal polynomial is orthogonal to all polynomials with a degree lower than that of itself.

**Lemma B.2.1.** *A polynomial $q(t) \in \mathbb{P}_n[t]$ satisfies $(q, r) = 0$ for all $r(t) \in \mathbb{P}_m[t]$ with $m < n$ if and only if $q(t) = p_n(t)$ up to some constant coefficient, where $p_n(t)$ denotes the orthogonal polynomial with degree $n$.*

*Proof.* $\implies$: Consider $q(t) = \alpha_n t^n + O(t^{n-1})$ and $p_n(t) = \tilde{\alpha}_n t^n + O(t^{n-1})$. Then we define

$$s(t) = q(t) - \frac{\alpha_n}{\tilde{\alpha}_n} p_n(t) = O(t^{n-1}),$$

which has a degree at most $n - 1$. Therefore, for all $m < n$,

$$(s, p_m) = (q, p_m) - \frac{\alpha_n}{\tilde{\alpha}_n}(p_n, p_m) = 0.$$

The former inner product $(q, p_m) = 0$ by assumption, while the latter inner product $(p_n, p_m) = 0$ by orthogonality. By Proposition B.1.1,

$$s(t) = \sum_{m=0}^{n-1} \frac{(p_m, s)}{(p_m, p_m)} p_m(t) = 0 \implies q(t) = \frac{\tilde{\alpha}_n}{\alpha_n} p_n(t).$$

$\impliedby$: Consider $r(t) = \sum_{k=0}^{m} r_k p_k(t)$. Let $q(t) = c p_n(t)$. Using the linearity of the inner product and orthogonality of $(p_n)_{n \in \mathbb{N}}$, for all $m < n$,

$$(q, r) = \left( c p_n(t), \sum_{k=0}^{m} r_k p_k(t) \right) = c \sum_{k=0}^{m} r_k (p_n(t), p_k(t)) = 0.$$

$\square$

Now, we have enough tools to prove the famous three-term recurrence relation.

*Proof of Theorem B.2.* Consider a sequence of orthogonal polynomials $(p_n)_{n \in \mathbb{N}}$. When $n = 1$, $p_1$ can be expressed as $A_1 t + B_1$ for $A_1, B_1 \in \mathbb{R}$. This is because $p_1$ is an element in an orthogonal basis with degree 1. Based on the inner product of orthogonal polynomials,

$$(p_k, t p_n) = \int t p_k(t) p_n(t) w(t) dt = (t p_k, p_n).$$

Therefore, for $0 \le k < n - 1$, we have $(p_k, t p_n) = 0$ by Lemma B.2.1. Since $t p_n(t)$ has degree $n + 1$, by Proposition B.1.1,

$$t p_n(t) = \sum_{k=0}^{n+1} \frac{(p_k, t p_n)}{(p_k, p_k)} p_k(t) = \sum_{k=n-1}^{n+1} \frac{(p_k, t p_n)}{(p_k, p_k)} p_k(t) = \alpha_{n-1} p_{n-1}(t) + \alpha_n p_n(t) + \alpha_{n+1} p_{n+1}(t)$$

$$\implies p_{n+1} = \left( \frac{1}{\alpha_{n+1}} t - \frac{\alpha_n}{\alpha_{n+1}} \right) p_n(t) - \frac{\alpha_{n-1}}{\alpha_{n+1}} p_{n-1}(t).$$

$\square$

**Remark.** *Recurrence is the core property of orthogonal polynomials in our setting, as one can find higher-order coefficients based on lower-order coefficients given the analytical form of the orthogonal polynomials. This idea coincides with the shuffle identity of signatures. As stated in Theorem 3.2, one can construct an explicit recurrence relation for the coefficients of orthogonal polynomials by linear functionals acting on signatures.*

APPROXIMATION RESULTS FOR FUNCTIONS IN $L_w^2$

Without loss of generality, consider $f \in L_w^2(-1, 1)$, as we can always transform an arbitrary interval $[a, b]$ linearly into the interval $[-1, 1]$. Recall the $N$-th degree approximation $P_N f(t)$ defined in Equation (19). The uniform convergence of the $N$-th degree approximation $P_N f(t)$ to $f$ can be found in Atkinson (2009), where we obtain

$$\frac{1}{\sqrt{2\pi}}\|f - P_N f\|_2 \leq \|f - P_N f\|_\infty \leq (1 + \|P_N\|)\|f - q\|_\infty, \qquad q \in \mathbb{P}_N,$$

where $\|P_N\|$ relates to the system of orthogonal polynomials, and $\|f - q\|_\infty$ depends on the smoothness of $f$. In the case of Chebyshev polynomials, where the weight function is $w(t) = 1/\sqrt{1 - t^2}$, $\|P_N\| = \frac{4}{\pi} \log n + \mathcal{O}(1)$ (Atkinson, 2009). For some $\alpha \in (0, 1]$,

$$\|f - P_N f\|_2 \leq c_k \frac{\log N}{N^{k+\alpha}} \qquad \text{for } N \geq 2.$$

This bound result is shown numerically in Figure 6.

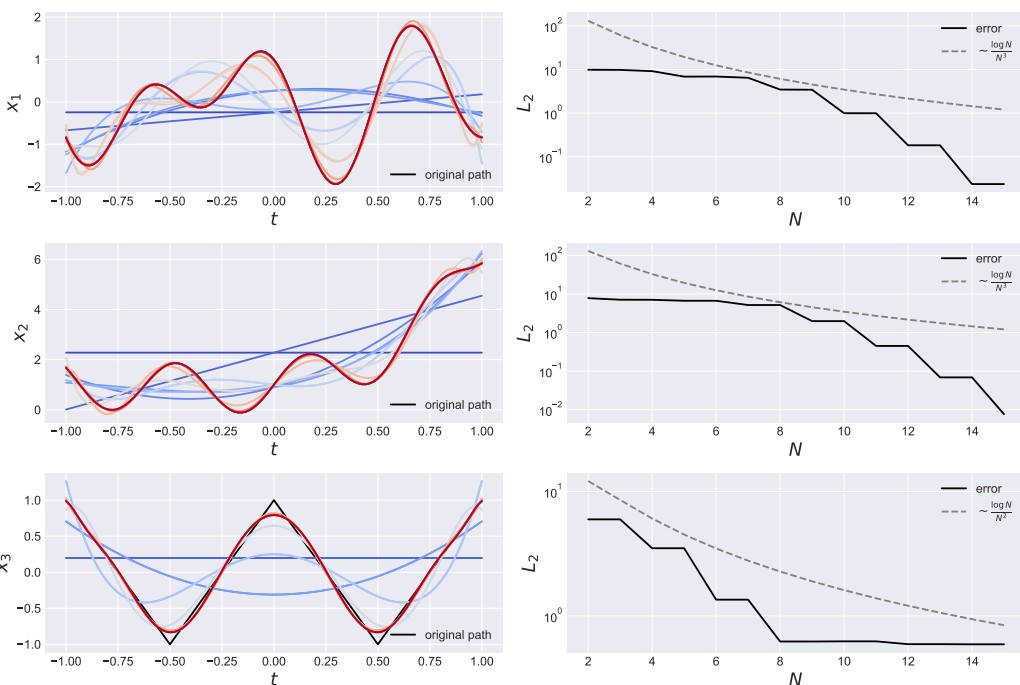

Figure 6: The approximation quality (left) and convergence of the $L_2$ error (right) for Chebyshev polynomials of increasing degree $N$. As $N$ increases, the colours change from blue to red in the left column. Paths are given by (top to bottom): $x_1(t) = \cos(10t) - \sin(2\pi t)$, $x_2(t) = \sin(10t) + e^{2t} - t$, $x_3(t) = 2|2t - 1| - 1$.

### B.1.3 EXAMPLES

In this subsection, we will provide two general orthogonal polynomial families, Jacobi polynomials and Hermite polynomials, which will be used for signature inversion in the next section. Figure 7 visualises the first few polynomials of these two kinds.

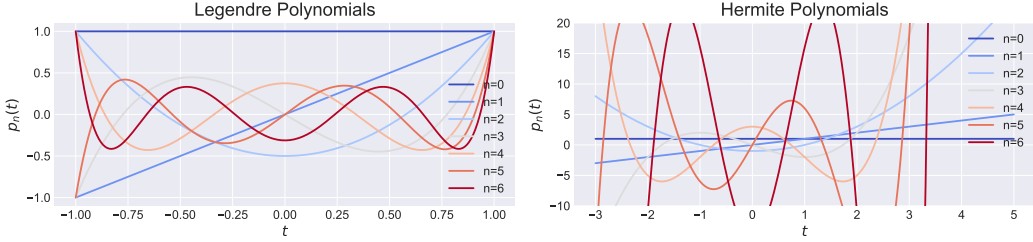

Figure 7: Visualisation of the first 7 Legendre and Hermite polynomials.

JACOBI POLYNOMIALS

Jacobi polynomials $p_n^{(\alpha,\beta)}$ are a system of orthogonal polynomials with respect to the weight function $w : (-1, 1) \to \mathbb{R}$ such that

$$w(t; \alpha, \beta) = (1 - t)^\alpha (1 + t)^\beta.$$

There are many well-known special cases of Jacobi polynomials, such as Legendre polynomials $p_n^{(0,0)}$ and Chebyshev polynomials $p_n^{(-1/2,-1/2)}$. In general, the analytical expression of Jacobi polynomials (Ismail, 2005) is defined by the hypergeometric function $_2F_1$:

$$p_n^{(\alpha,\beta)}(t) = \frac{(\alpha + 1)_n}{n!} {}_2F_1(-n, 1 + \alpha + \beta + n; \alpha + 1; \frac{1}{2}(1 - t)),$$

where $(\alpha + 1)_n$ is the Pochhammer's symbol. For orthogonality, Jacobi polynomials satisfy

$$\int_{-1}^{1} (1-t)^\alpha (1+t)^\beta p_m^{(\alpha,\beta)}(t) p_n^{(\alpha,\beta)}(t) = \frac{2^{\alpha+\beta+1}\Gamma(n+\alpha+1)\Gamma(n+\beta+1)}{(2n+\alpha+\beta+1)\Gamma(n+\alpha+\beta+1)n!}\delta_{nm}, \qquad \alpha, \beta > -1,$$

where $\delta_{mn}$ is the Kronecker delta. For fixed $\alpha, \beta$, the recurrence relation of Jacobi polynomials is

$$p_n^{(\alpha,\beta)}(t) = \frac{2n + \alpha + \beta - 1}{2n(n + \alpha + \beta)(2n + \alpha + \beta - 2)} \left((2n + \alpha + \beta)(2n + \alpha + \beta - 2)t + \alpha^2 - \beta^2\right) p_{n-1}^{(\alpha,\beta)}(t)$$
$$- \frac{(n + \alpha - 1)(n + \beta - 1)(2n + \alpha + \beta)}{n(n + \alpha + \beta)(2n + \alpha + \beta - 2)} p_{n-2}^{(\alpha,\beta)}(t).$$

HERMITE POLYNOMIALS

Hermite polynomials are a system of orthogonal polynomials with respect to the weight function $w : (-\infty, \infty) \to \mathbb{R}$ such that $w(t) = \exp(-t^2/2)$. These are called the probabilist's Hermite polynomials, which we will use throughout the section. There is another form called the physicist's Hermite polynomials with respect to the weight function $w(t) = \exp(-t^2)$. The explicit expression of the probabilist's Hermite polynomials can be written as

$$H_n(t) = n! \sum_{m=0}^{\lfloor \frac{n}{2} \rfloor} \left(-\frac{1}{2}\right)^m \frac{t^{n-2m}}{m!(n - 2m)!},$$

with the orthogonality property

$$\int_{-\infty}^{\infty} H_m(t) H_n(t) e^{-\frac{t^2}{2}} dt = \sqrt{2\pi} n! \delta_{mn}. \tag{20}$$

Lastly, we state the recurrence relation of Hermite polynomials as $H_{n+1}(t) = xH_n(t) - nH_{n-1}(t)$. Note that the weight of Hermite polynomials can be viewed as an unnormalised normal distribution. If we are more interested in a particular region far away from the origin, we can define a "shift-and-scale" version of Hermite polynomials with respect to the weight

$$w^{t_0,\epsilon}(t) = \exp((t - t_0)^2/2\epsilon^2),$$

where $t_0$ denotes the new centre, and $\epsilon$ is the standard deviation. Let $(H_n^{t_0,\epsilon})_{n\in\mathbb{N}}$ denote the shift-and-scale Hermite polynomials. Then, the orthogonality property is

$$\int_{-\infty}^{\infty} H_m^{t_0,\epsilon}(t) H_n^{t_0,\epsilon}(t) e^{-\frac{(t-t_0)^2}{2\epsilon^2}} dt = \epsilon \int_{-\infty}^{\infty} H_m^{t_0,\epsilon}(t_0 + \epsilon y) H_n^{t_0,\epsilon}(t_0 + \epsilon y) e^{-\frac{y^2}{2}} dy,$$

by substitution $y = (t - t_0)/\epsilon$. Hence, if

$$H_n^{t_0,\epsilon}(t_0 + \epsilon y) = H_n(y), \qquad n \in \mathbb{N}, \tag{21}$$

then $(H_n^{t_0,\epsilon})_{n\in\mathbb{N}}$ is an orthogonal polynomial system with orthogonality

$$\int_{-\infty}^{\infty} H_m^{t_0,\epsilon}(t) H_n^{t_0,\epsilon}(t) e^{-\frac{(t-t_0)^2}{2\epsilon^2}} dt = \epsilon \int_{-\infty}^{\infty} H_m(y) H_n(y) e^{-\frac{y^2}{2}} dy = \epsilon\sqrt{2\pi} n! \delta_{mn},$$

which follows from the orthogonality of Hermite polynomials in Equation (20). Similarly, the connection between Hermite and shift-and-scale Hermite polynomials in Equation (21) provides a way to find the explicit form and recurrence relation of $(H_n^{t_0,\epsilon})_{n\in\mathbb{N}}$, which are

$$H_n^{t_0,\epsilon}(t) = n! \sum_{m=0}^{\lfloor\frac{n}{2}\rfloor} \left(-\frac{1}{2}\right)^m \frac{1}{m!(n-2m)!} \left(\frac{t-t_0}{\epsilon}\right)^{n-2m}, \tag{22}$$

$$H_{n+1}^{t_0,\epsilon}(t) = \frac{1}{\epsilon}(t-t_0) H_n^{t_0,\epsilon}(t) - n H_{n-1}^{t_0,\epsilon}(t). \tag{23}$$

**Remark.** *Note that there is a simple expression for* $(H_n^{t_0,\epsilon})_{n\in\mathbb{N}}$ *at* $t = t_0$. *One can easily observe that*

$$H_n^{t_0,\epsilon}(t_0) = \begin{cases} \left(-\frac{1}{2}\right)^{\frac{n}{2}} \frac{n!}{\frac{n}{2}!} & \text{for even } n \\ 0 & \text{for odd } n \end{cases}.$$

## B.2 FOURIER SERIES

One can also represent a function by a trigonometric series. Here, we only present a brief introduction to the Fourier series, providing complementary details to the main result in Theorem 3.1.

### B.2.1 TRIGONOMETRIC SERIES

Let $f \in L^1(-\pi, \pi)$. The Fourier series of $f$ is defined by

$$F(t) = \frac{a_0}{2} + \sum_{k=1}^{\infty} (a_k \cos(kt) + b_k \sin(kt)),$$

where

$$a_k = \frac{1}{\pi} \int_{-\pi}^{\pi} f(t) \cos(kt) dx, \qquad k \geq 0$$

$$b_k = \frac{1}{\pi} \int_{-\pi}^{\pi} f(t) \sin(kt) dx, \qquad k \geq 1,$$

which can be derived from the orthogonal bases $\{\cos kt\}_k$ and $\{\sin kt\}_k$. More generally, we can extend the period to $2l \in \mathbb{R}$. For $f \in L^1(-l, l)$ and $k \in \mathbb{Z}$,

$$F(t) = \sum_{n=-\infty}^{\infty} c_k e^{i\frac{2\pi}{l}kt}, \qquad c_k = \frac{1}{l} \int_0^l f(t) e^{-i\frac{2\pi}{l}kt} dt. \tag{24}$$

In the setting of the Fourier series, the expression for the $k$-th coefficient $c_k$ in the exponential form can be defined as a linear functional $\mathcal{L}_k(x) = c_k^x$ on the space of Fourier series, for $x \in L^1(-l, l)$.

### B.2.2 CONVERGENCE

Under some regularity conditions, $F(t)$ converges to $f(t)$ (Atkinson, 2009). To examine the convergence of the Fourier series, we define the partial sum of the Fourier series as

$$S_N f(t) = \frac{a_0}{2} + \sum_{k=1}^{N} (a_k \cos(kt) + b_k \sin(kt)).$$

Now we present pointwise convergence and uniform convergence results (Atkinson, 2009) of the Fourier series for various functions.

**Theorem B.3** (Pointwise convergence for bounded variation). *For a $2\pi$-periodic function $f$ of bounded variation on $[-\pi, \pi]$, its Fourier series at an arbitrary $t$ converges to*

$$\frac{1}{2} \left( f(t^-) + f(t^+) \right).$$

**Theorem B.4** (Uniform convergence for piecewise smooth functions). *If $f$ is a $2\pi$-periodic piecewise smooth function,*

(a) *if $f$ is also continuous, then the Fourier series converges uniformly and continuously to $f$;*

(b) *if $f$ is not continuous, then the Fourier series converges uniformly to $f$ on every closed interval without discontinuous points.*

**Theorem B.5** (Uniform error bounds). *Let $f \in C_p^{k,\alpha}(2\pi)$ be a $2\pi$-periodic $k$-times continuously differentiable function that is Hölder continuous with the exponent $\alpha \in (0, 1]$. Then, the 2-norm and infinity-norm bound of the partial sum $S_N f$ can be expressed as*

$$\frac{1}{\sqrt{2\pi}} \|f - S_N f\|_2 \leq \|f - S_N f\|_\infty \leq c_k \frac{\log N}{N^{k+\alpha}}, \qquad for\ N \geq 2.$$

For functions only defined on an interval $[a, b]$, we can always shift and extend them to be $2\pi$-periodic functions. These theorems guarantee the convergence of common functions we will use in later experiments. To illustrate this, Figure 8 compares the convergence theorem bounds to numerical approximation results. Note that compared with the path $x_2(t) = \sin(10t) + e^{2t} - t$, the other 2 paths have better empirical convergence results. The main reason is that the Fourier series of $x_2$ at $t = \pm 1$ does not pointwise converge to $x_2(\pm 1)$. Since the Fourier series treats the interval $[-1, 1]$ as one period over $\mathbb{R}$, by Theorem B.3, the series will converge to $(x_2(-1) + x_2(1))/2$ at $t = \pm 1$, leading to incorrect convergence at boundaries. This property also arises in real-world non-periodic time series, leading us to introduce the *mirror augmentation* later in Appendix E.

Comparing Figures 6 and 8, one can observe that orthogonal polynomials are better at approximating continuously differentiable paths, while the Fourier series is better at estimating paths with spikes, and its computation is more stable in the long run. In a later section, Figure 9 provides a summary of convergence results for different types of orthogonal polynomials and Fourier series, which also match the results shown here.

### B.3 Approximation quality of orthogonal polynomials and Fourier series

Finally, we present a numerical comparison of the approximation results given by the methods introduced above.

In particular, we will visualise the approximation results of the following objects:

- Legendre polynomials: $w(t) = 1$
- two types of Jacobi polynomials: $w(t) = \sqrt{1+t}$, $w(t) = \sqrt{1-t}$
- three types of shift-and-scale Hermite polynomials with different variance for *pointwise approximation*: $\epsilon = 0.1, 0.05, 0.01$
- Fourier series

For pointwise approximation via the Hermite polynomials, each sample point $t_i$ of the function will be approximated by a system of Hermite polynomials centred at the point $t_i$, i.e., $(H_n^{t_i, \epsilon})_{n \in \mathbb{N}}$. To test the approximation quality, we simulate random polynomial functions and random trigonometric functions. The $L_2$ error is then obtained by an average over 10 functions of each type.

### B.3.1 Approximation results

Figure 9 demonstrates the reduction in $L_2$ error as the order of orthogonal polynomials and Fourier series increases. Among all the bases considered, the Fourier series provides the least accurate approximation for both path types. This is due to its inability to guarantee pointwise convergence at $\pm 1$, which is caused by boundary inconsistencies. Three types of Jacobi polynomials, including

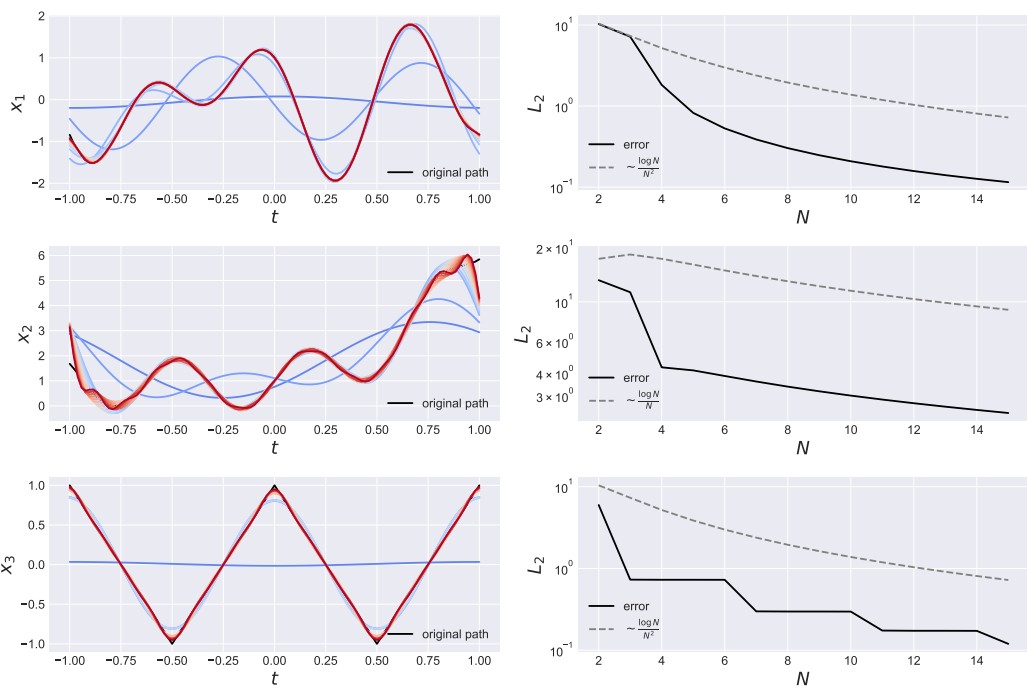

Figure 8: Approximation (left) and $L_2$ convergence (right) results for Fourier series by increasing order $N$, with the same experimental setting as Figure 6.

Legendre polynomials, show comparable approximation results, with a slight edge in convergence observed for Legendre polynomials. On the other hand, Hermite polynomials exhibit a much lower approximation error due to their shifting focus on the point of interest. However, decreasing $\epsilon$ to sharpen the focus on sample points can cause the coefficients of Hermite polynomials to inflate rapidly. This behaviour is consistent with the analytic form and recurrence relation of the shift-and-scale Hermite polynomials, as detailed in Equation (22) and Equation (23). As a result, Hermite polynomials with $\epsilon = 0.01$ do not outperform those with $\epsilon = 0.05$. The step-like pattern of decrease observed in Hermite polynomials can be traced back to Remark B.1.3. Figure 10 shows the $L_2$ approximation error of different bases for two of our real-world datasets. Here, we see that apart from using shift-and-scale Hermite polynomials which come with additional complexity (see Figure 3), the Fourier series seems to be the best candidate for approximation.

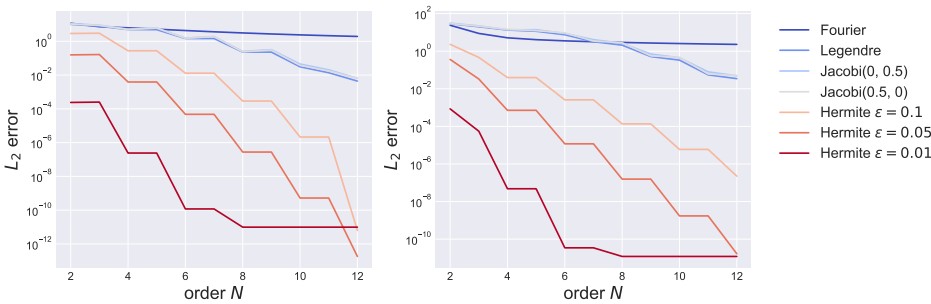

Figure 9: $L_2$ approximation error using different bases. The figures (from left to right) are the corresponding error averaged over 10 random polynomial and trigonometric functions.

To mitigate computational expense, we henceforth use Hermite polynomials with $\epsilon = 0.05$ as the representative of the Hermite family. The findings presented in Figures 9 and 10 play a crucial role

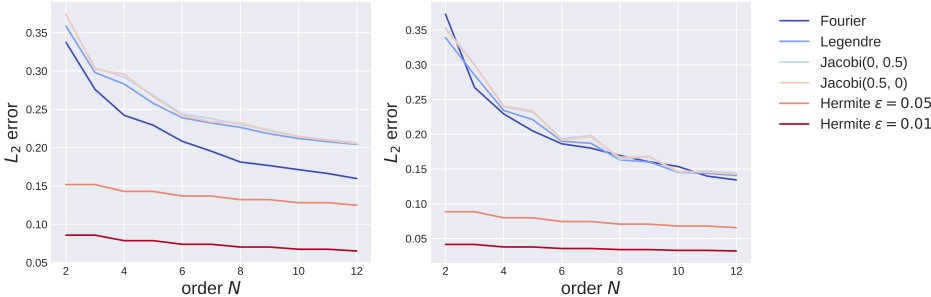

Figure 10: Real data $L_2$ approximation error using different bases. The figures (from left to right) are the corresponding error averaged over 10 random samples from the HEPC and Exchange rates datasets.

in our signature inversion method, as they establish a benchmark for the best possible performance attainable in path reconstruction from signatures.

## C    PROOFS OF SIGNATURE INVERSION

In this section, we present the formal proofs of the signature inversion Theorem 3.2 and Theorem 3.1, along with the remark in Section 3.2 about Taylor approximation of the weight function.

### C.1    PROOF OF ORTHOGONAL POLYNOMIAL INVERSION THEOREM 3.2

Recall the statement in Theorem 3.2 deriving the $n$-th polynomial coefficient $\alpha_n$ (see Equation (10)) via a recurrence relation:

Let $x : [a, b] \to \mathbb{R}$ be a smooth path such that $x(a) = 0$. Consider the augmentation $\hat{x}(t) = (t, \omega(t)x(t)) \in \mathbb{R}^2$, where $\omega(t)$ corresponds to the weight function of a system of orthogonal polynomials $(p_n)_{n \in \mathbb{N}}$, and is well defined on the closed and compact interval $[a, b]$. Then, there exists a linear combination $\ell_n$ of words such that the $n^{th}$ coefficient in Equation (10) satisfies $\alpha_n = \langle \ell_n, S(\hat{x}) \rangle$. Furthermore, the sequence $(\ell_n)_{n \in \mathbb{N}}$ satisfies the following recurrence relation

$$\ell_n = A_n \frac{(p_{n-1}, p_{n-1})}{(p_n, p_n)} e_1 \succ \ell_{n-1} + (A_n a + B_n) \frac{(p_{n-1}, p_{n-1})}{(p_n, p_n)} \ell_{n-1} + C_n \frac{(p_{n-2}, p_{n-2})}{(p_n, p_n)} \ell_{n-2},$$

with

$$\ell_0 = \frac{A_0}{(p_0, p_0)} e_{21} \quad \text{and} \quad \ell_1 = \frac{A_1}{(p_1, p_1)} (e_{121} + e_{211}) + \frac{A_1 a + B_1}{(p_1, p_1)} e_{21}.$$

*Proof.* One can express the first two coefficients in an orthogonal polynomial expansion of $x$ by the signature:

$$\alpha_0 = \frac{1}{(p_0, p_0)} \int_a^b A_0 x(t)\omega(t)dt$$

$$= \langle \frac{A_0}{(p_0, p_0)} e_2 \succ e_1, S(\hat{x}) \rangle$$

$$= \langle \frac{A_0}{(p_0, p_0)} e_{21}, S(\hat{x}) \rangle$$

$$= \langle \ell_0, S(\hat{x}) \rangle,$$

$$\alpha_1 = \frac{1}{(p_1, p_1)} \int_a^b (A_1 t + B_1) x(t)\omega(t)dt$$

$$= \frac{A_1}{(p_1, p_1)} \int_a^b (t - a) x(t)\omega(t)dt + \frac{A_1 a + B_1}{(p_1, p_1)} \int_a^b x(t)\omega(t)dt$$

$$= \frac{A_1}{(p_1, p_1)} \langle (e_1 \sqcup\!\sqcup e_2) \succ e_1, S(\hat{x}) \rangle + \frac{A_1 a + B_1}{(p_1, p_1)} \langle e_{21}, S(\hat{x}) \rangle$$

$$= \langle \frac{A_1}{(p_1, p_1)} (e_{121} + e_{211}) + \frac{A_1 a + B_1}{(p_1, p_1)} e_{21}, S(\hat{x}) \rangle$$

$$= \langle \ell_1, S(\hat{x}) \rangle.$$

Then one can find $\ell_n$ recursively by multiplying both sides of Equation (9) by $x(t)\omega(t)$ and integrating on $[a, b]$:

$$\int_a^b p_n(t)x(t)\omega(t)dt = \int_a^b (A_n t + B_n)p_{n-1}(t)x(t)\omega(t)dt + \int_a^b C_n p_{n-2}(x)x(t)\omega(t)dt$$

$$= A_n \int_a^b (t - a)d\left(\int_a^t p_{n-1}(s)x(s)\omega(s)ds\right)$$

$$+ (A_n a + B_n) \int_a^b p_{n-1}(t)x(t)\omega(t)dt$$

$$+ C_n \int_a^b p_{n-2}(x)x(t)\omega(t)dt.$$

By definition of $\alpha_n$,

$$\int_a^b p_n(t)x(t)\omega(t)d = (p_n, p_n)\alpha_n = (p_n, p_n)\langle \ell_n, S(\hat{x}) \rangle,$$

$$A_n \int_a^b (t - a)d\left(\int_a^t p_{n-1}(s)x(s)\omega(s)ds\right) = A_n (p_{n-1}, p_{n-1})\langle e_1 \succ \ell_{n-1}, S(\hat{x}) \rangle,$$

$$(A_n a + B_n) \int_a^b p_{n-1}(t)x(t)\omega(t)dt = (A_n a + B_n)(p_{n-1}, p_{n-1})\langle \ell_{n-1}, S(\hat{x}) \rangle,$$

$$C_n \int_a^b p_{n-2}(x)x(t)\omega(t)dt = C_n (p_{n-2}, p_{n-2})\langle \ell_{n-2}, S(\hat{x}) \rangle.$$

Therefore, the recurrence relation of the linear functions on the signature retrieving the coefficients of orthogonal polynomials is

$$\langle \ell_n, S(\hat{x}) \rangle = A_n \frac{(p_{n-1}, p_{n-1})}{(p_n, p_n)} \langle e_1 \succ \ell_{n-1}, S(\hat{x}) \rangle$$

$$+ (A_n a + B_n) \frac{(p_{n-1}, p_{n-1})}{(p_n, p_n)} \langle \ell_{n-1}, S(\hat{x}) \rangle$$

$$+ C_n \frac{(p_{n-2}, p_{n-2})}{(p_n, p_n)} \langle \ell_{n-2}, S(\hat{x}) \rangle.$$

$\square$

Several assumptions are made in order to derive the recurrence relation. Namely, the interval defined on the inner space must be compact, and the weight function $\omega(t)$ should be well-defined on the closed interval $[a, b]$. These assumptions can limit the range of applicable orthogonal polynomial families. For example, since the range of Hermite polynomials is unbounded, they are not suitable for our inversion. However, by using a shift-and-scale version of these polynomials, where most of the weight density is concentrated at a particular point, their weight can be numerically truncated to a compact interval. One can centre the weight density using a small enough $\epsilon$ and shift it to a point of interest $t_i$. Since the non-zero density region is now concentrated in a small interval, Theorem 3.2 can be applied to the truncated density over this interval. The relationship between the original Hermite polynomials and the shift-and-scale Hermite polynomials is given by Equation (21).

## C.2  REMARK ON THE TAYLOR APPROXIMATION OF THE WEIGHT FUNCTION

The results in Theorem 3.2 require signatures of $\hat{x} = (t, w(t)x(t))$. However, sometimes one may only have signatures of $\tilde{x} = (t, x(t))$. Here, we propose a theoretically applicable method by approximating the weight function as a Taylor polynomial.

Consider the Taylor approximation of $\omega$ around $t = a$, i.e.,

$$\omega(t) \approx \sum_{i=0}^{M} \frac{d^i \omega}{dt^i}\bigg|_{t=a} (t-a)^i = \sum_{i=0}^{M} \omega_i (t-a)^i.$$

Letting $\tilde{x}_t = (t, x(t))$ and

$$c_i := (e_2 \sqcup\!\sqcup e_1^{\sqcup\!\sqcup i}) \succ e_1 = i!(e_{21\ldots 1} + e_{121\ldots 1} + \ldots + e_{1\ldots 121}),$$

we have

$$\alpha_0 = \frac{1}{(p_0, p_0)} \int_a^b A_0 x(t)\omega(t) dt$$

$$= \frac{A_0}{(p_0, p_0)} \sum_{i=0}^{M} \omega_i \int_a^b (t-a)^i x(t) dt$$

$$= \langle \frac{A_0}{(p_0, p_0)} \sum_{i=0}^{M} \omega_i (e_2 \sqcup\!\sqcup e_1^{\sqcup\!\sqcup i}) \succ e_1, S(\tilde{x}) \rangle$$

$$= \langle \frac{A_0}{(p_0, p_0)} \sum_{i=0}^{M} \omega_i c_i, S(\tilde{x}) \rangle$$

$$= \langle \ell_0, S(\tilde{x}) \rangle,$$

$$\alpha_1 = \frac{1}{(p_1, p_1)} \int_a^b (A_1 t + B_1) x(t)\omega(t) dt$$

$$= \frac{1}{(p_1, p_1)} \int_a^b \left( A_1(t-a) + A_1 a + B_1 \right) x(t)\omega(t) dt$$

$$= \frac{1}{(p_1, p_1)} \sum_{i=0}^{M} \omega_i \int_a^b \left( A_1(t-a)^{i+1} + (A_1 a + B_1)(t-a)^i \right) x(t) dt$$

$$= \langle \frac{1}{(p_1, p_1)} \sum_{i=0}^{M} \omega_i \left( A_1(e_2 \sqcup\!\sqcup e_1^{\sqcup\!\sqcup i+1}) + (A_1 a + B_1)(e_2 \sqcup\!\sqcup e_1^{\sqcup\!\sqcup i}) \right) \succ e_1, S(\tilde{x}) \rangle$$

$$= \langle \frac{1}{(p_1, p_1)} \sum_{i=0}^{M} \omega_i (A_1 c_{i+1} + (A_1 a + B_1) c_i), S(\tilde{x})_{a,b} \rangle$$

$$= \langle \ell_1, S(\tilde{x}) \rangle$$

By induction, the same relation as Theorem 3.2 holds

$$\ell_n = A_n \frac{(p_{n-1}, p_{n-1})}{(p_n, p_n)} e_1 \succ \ell_{n-1} + (A_n a + B_n) \frac{(p_{n-1}, p_{n-1})}{(p_n, p_n)} \ell_{n-1} + C_n \frac{(p_{n-2}, p_{n-2})}{(p_n, p_n)} \ell_{n-2}.$$

There are several reasons why we consider the Taylor approximation method "theoretically applicable." The expansion of the weight function around a point $a$ can be hard to find analytically, and even if the series is found, it may diverge. If the series does converge, we still need to determine the number of terms required to meet a certain error tolerance. Furthermore, if the convergence rate is slow, more terms are needed in the series. This leads to higher necessary signature truncation levels and consequently also higher computational complexity.

### C.3 PROOF OF FOURIER INVERSION IN THEOREM 3.1

Recall the statement in Theorem 3.1 deriving the Fourier coefficients $a_0, a_n, b_n$ (see Equations (5), (6), (7)) of a path as follows:

Let $x : [0, 2\pi] \to \mathbb{R}$ be a periodic smooth path such that $x(0) = 0$, and consider the augmentation $\hat{x}(t) = (t, \sin(t), \cos(t) - 1, x(t)) \in \mathbb{R}^4$. Then the following relations hold

$$a_0 = \frac{1}{2\pi} \langle e_4 \succ e_1, S(\hat{x}) \rangle,$$

$$a_n = \frac{1}{\pi} \sum_{k=0}^{n} \sum_{q=0}^{k} \binom{n}{k} \binom{k}{q} \cos(\frac{1}{2}(n-k)\pi) \langle (e_4 \shuffle e_2^{\shuffle n-k} \shuffle e_3^{\shuffle q}) \succ e_1, S(\hat{x}) \rangle,$$

$$b_n = \frac{1}{\pi} \sum_{k=0}^{n} \sum_{q=0}^{k} \binom{n}{k} \binom{k}{q} \sin(\frac{1}{2}(n-k)\pi) \langle (e_4 \shuffle e_2^{\shuffle n-k} \shuffle e_3^{\shuffle q}) \succ e_1, S(\hat{x}) \rangle.$$

*Proof.* By Multiple-Angle formulas, we have

$$\sin(nt) = \sum_{k=0}^{n} \binom{n}{k} \cos^k(t) \sin^{n-k}(t) \sin(\frac{1}{2}(n-k)\pi), \tag{25}$$

$$\cos(nt) = \sum_{k=0}^{n} \binom{n}{k} \cos^k(t) \sin^{n-k}(t) \cos(\frac{1}{2}(n-k)\pi). \tag{26}$$

We can now connect Equation (7), Equation (25), and the shuffle identity of the signature described in Lemma A.0.1 to obtain an expression for $b_n$ as

$$
\begin{aligned}
b_n &= \frac{1}{\pi} \int_0^{2\pi} x(t) \sin(nt) dt \\
&= \frac{1}{\pi} \int_0^{2\pi} x(t) \sum_{k=0}^{n} \binom{n}{k} \cos^k(t) \sin^{n-k}(t) \sin(\frac{1}{2}(n-k)\pi) dt \\
&= \frac{1}{\pi} \sum_{k=0}^{n} \binom{n}{k} \sin(\frac{1}{2}(n-k)\pi) \int_0^{2\pi} x(t) \cos^k(t) \sin^{n-k}(t) dt \\
&= \frac{1}{\pi} \sum_{k=0}^{n} \binom{n}{k} \sin(\frac{1}{2}(n-k)\pi) \int_0^{2\pi} x(t)((\cos(t)-1)+1)^k \sin^{n-k}(t) dt \\
&= \frac{1}{\pi} \sum_{k=0}^{n} \binom{n}{k} \sin(\frac{1}{2}(n-k)\pi) \int_0^{2\pi} x(t) \sin^{n-k}(t) \sum_{q=0}^{k} \binom{k}{q} (\cos(t)-1)^q dt \\
&= \frac{1}{\pi} \sum_{k=0}^{n} \sum_{q=0}^{k} \binom{n}{k} \binom{k}{q} \sin(\frac{1}{2}(n-k)\pi) \int_0^{2\pi} x(t) \sin^{n-k}(t)(\cos(t)-1)^q dt \\
&= \frac{1}{\pi} \sum_{k=0}^{n} \sum_{q=0}^{k} \binom{n}{k} \binom{k}{q} \sin(\frac{1}{2}(n-k)\pi) \langle (e_4 \shuffle e_2^{\shuffle n-k} \shuffle e_3^{\shuffle q}) \succ e_1, S(\hat{x}) \rangle.
\end{aligned}
\tag{27}
$$

Table 3: Polynomial/Fourier basis orders used in Figures 11 and 12.

| Approximation method | Order $n$ | Level of truncated signature |
|---|---|---|
| Legendre | 10 | n+2=12 |
| Jacobi$(0, 0.5)$ | 10 | n+2=12 |
| Jacobi$(0.5, 0)$ | 10 | n+2=12 |
| Hermite ($\epsilon = 0.05$) | 2 | n+2=4 |
| Fourier | 10 | n+2=12 |

Similarly, we rearrange Equation (6) with Equation (26) to obtain the formula for $a_n$

$$
\begin{aligned}
a_n &= \frac{1}{\pi} \int_0^{2\pi} x(t) \cos(nt) dt \\
&= \frac{1}{\pi} \sum_{k=0}^{n} \sum_{q=0}^{k} \binom{n}{k} \binom{k}{q} \cos(\frac{1}{2}(n-k)\pi) \langle (e_4 \shuffle e_2^{\shuffle n-k} \shuffle e_3^{\shuffle q}) \succ e_1, S(\hat{x}) \rangle.
\end{aligned}
\tag{28}
$$

Finally, we get $a_0$ immediately as

$$
a_0 = \frac{1}{2\pi} \int_0^{2\pi} x(t) dt = \langle e_4 \succ e_1, S(\hat{x}) \rangle.
\tag{29}
$$

$\square$

## D    VISUALISING INVERSION BY DIFFERENT BASES

We demonstrate the quality of signature inversion on paths generated from fractional Brownian motion (Mandelbrot & Van Ness, 1968) (see Figure 11), and two of our real-world datasets (see Figure 12) using five different polynomial and Fourier bases. The orders of the bases recovered in these figures are listed in Table D, establishing a relationship with the levels of truncated signatures. The reconstruction results are influenced by three main factors:

1. the degree/order of the bases, $n$, and the corresponding levels of truncated signatures;
2. the complexity of the paths, such as frequency and smoothness;
3. the weight function of the orthogonal polynomials.

Factor 1 plays a crucial role in path reconstruction by providing a theoretical bound on the inversion error, as the order of the retrieved polynomial and Fourier coefficients is constrained by the truncation level of the signature. Recall Figure 9, which demonstrates that increasing the basis order improves the approximation quality, implying that reconstructions from higher-level signatures will more closely approximate the original path.

Factor 2 also plays a crucial role in the quality of approximation. A comparison between the first and second columns of Figure 11 shows that all bases can approximate the simpler path in the second column more accurately. Consequently, more complex paths may lead to less satisfactory inversion results due to the limitations of the bases, as previously discussed in Section 5.2.1.

Compared to the previous factors, factor 3 has a relatively minor impact on the reconstruction process. As shown in Figure 12, the left tail of the Jacobi$(0, 0.5)$ approximation and the right tail of the Jacobi$(0.5, 0)$ approximation tend to diverge as their weight functions approach zero at $t \to \pm 1$. Meanwhile, the signature inversion using Hermite polynomials, performed on a pointwise basis, remains precise even at lower polynomial degrees, as each sample point is estimated at the centre of the weight function.

## E    EXPERIMENT DETAILS

In this section, we provide additional details about the experimental setup. The accompanying code can be found at `https://github.com/Barb0ra/SigDiffusions`.

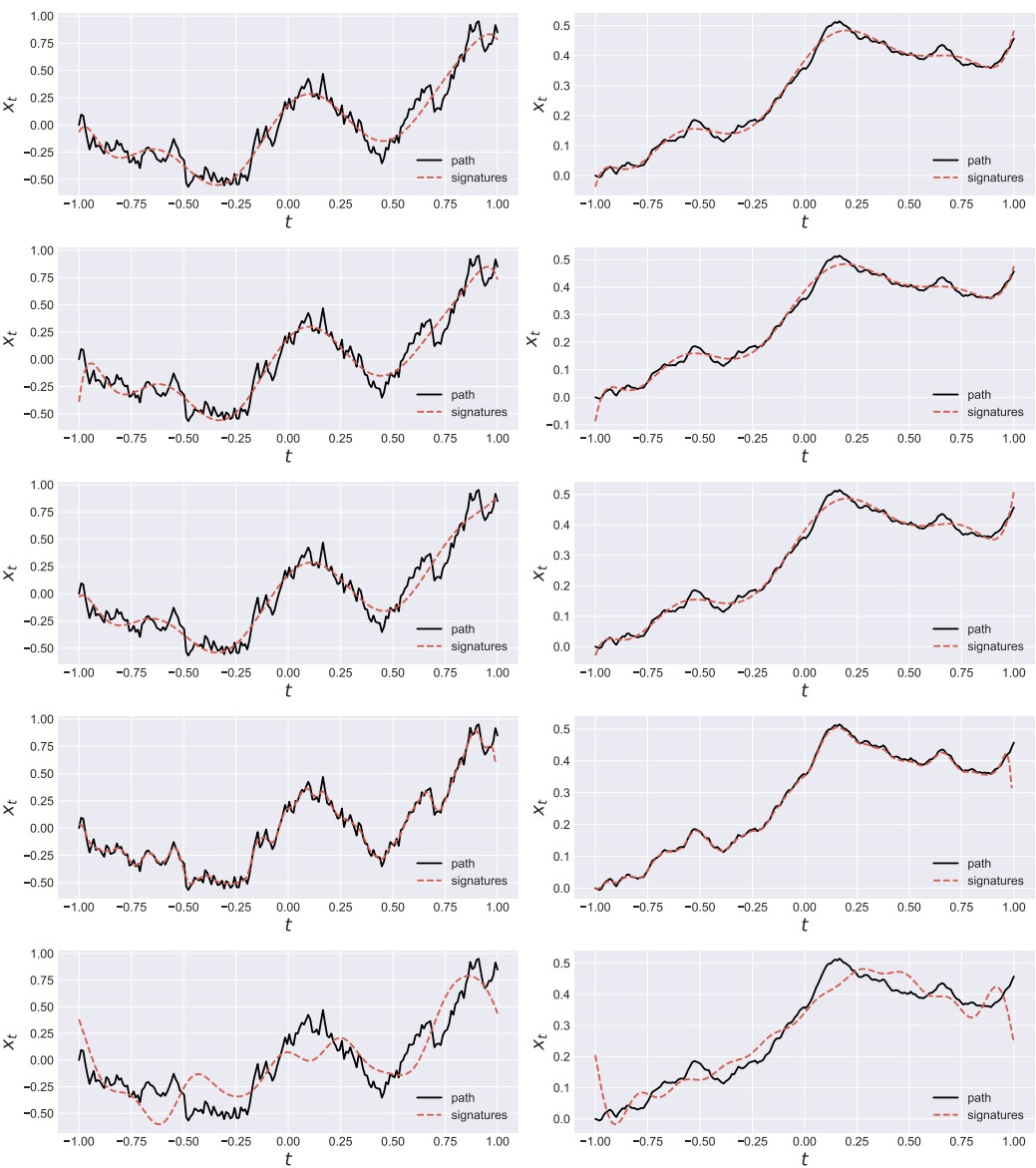

Figure 11: Inversion results on fractional Brownian motion with Hurst $0.5$ and $0.9$, with approximation bases (from top to bottom) Legendre (Jacobi$(0, 0)$), Jacobi$(0, 0.5)$, Jacobi$(0.5, 0)$, Hermite ($\epsilon = 0.05$) and Fourier.

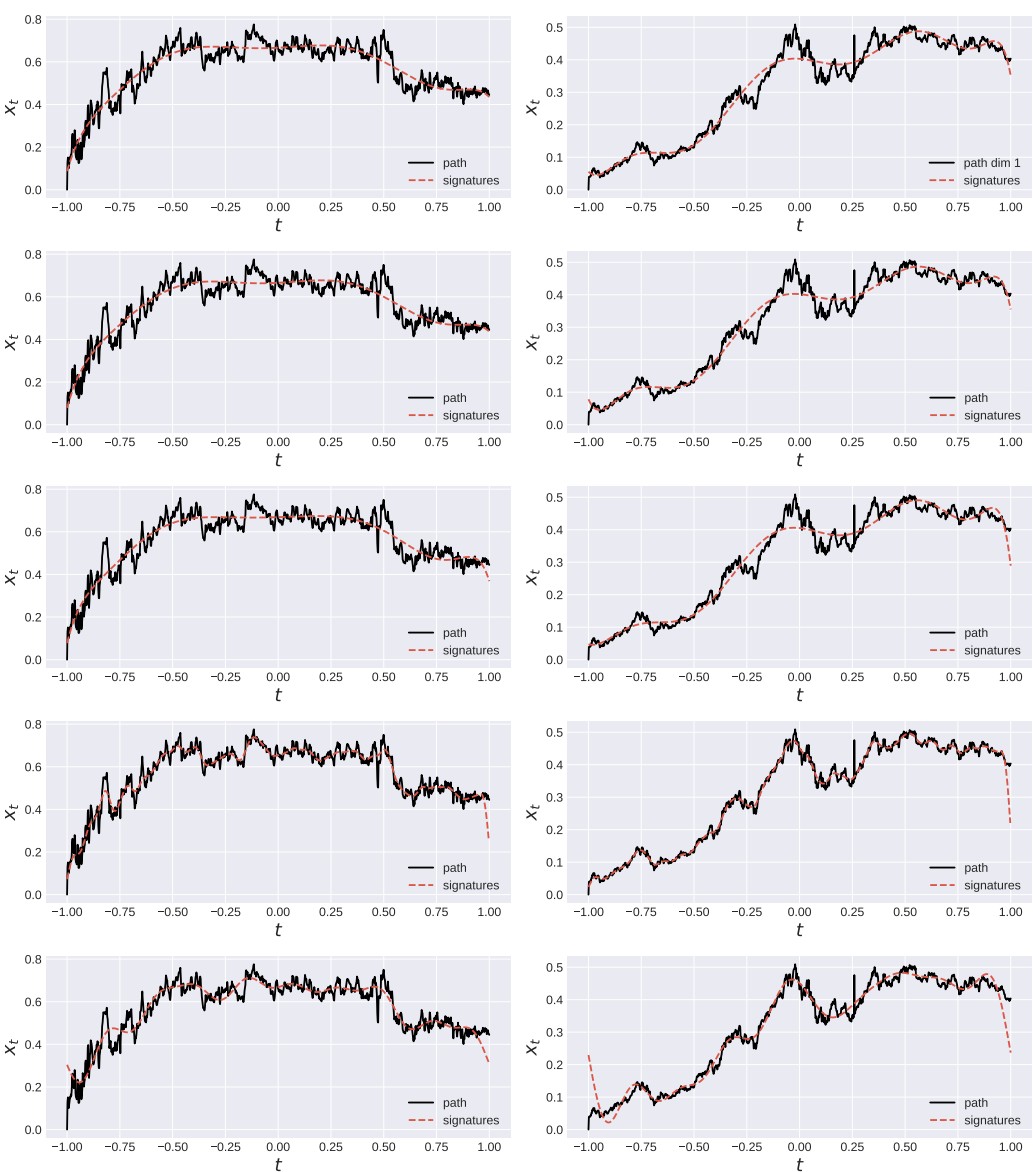

Figure 12: Inversion results on real-world time series. The left column is a sample from the HEPC dataset. The right column is a sample from the Exchange rates dataset (for readability, we only plot one of the eight dimensions). The approximation bases (from top to bottom) are Legendre (Jacobi$(0,0)$), Jacobi$(0, 0.5)$, Jacobi$(0.5, 0)$, Hermite ($\epsilon = 0.05$) and Fourier.

Table 4: KS Test average scores and type I errors on the marginals of time series of length 1000.

| Dataset | Model | t=300 | t=500 | t=700 | t=900 |
|---|---|---|---|---|---|
| Sines | SigDiffusion (ours) | 0.25, 50% | 0.22, 31% | 0.24, 40% | 0.23, 38% |
| | DDO ($\gamma = 1$) | **0.17, 7%** | **0.16, 5%** | **0.20, 15%** | **0.23, 33%** |
| | Diffusion-TS | 0.77, 100% | 0.50, 100% | 0.48, 100% | 0.48, 100% |
| | CSPD-GP (RNN) | 0.62, 100% | 0.45, 82% | 0.48, 95% | 0.55, 99% |
| | CSPD-GP (Transformer) | 0.57, 100% | 0.49, 100% | 0.51, 99% | 0.60, 100% |
| Predator-prey | SigDiffusion (ours) | **0.20, 20%** | 0.29, 76% | **0.23, 39%** | **0.22, 35%** |
| | DDO ($\gamma = 10$) | 0.34, 92% | 0.30, 85% | 0.36, 96% | 0.40, 100% |
| | Diffusion-TS | 1.00, 100% | 1.00, 100% | 1.00, 100% | 0.99, 100% |
| | CSPD-GP (RNN) | 0.27, 56% | **0.25, 47%** | 0.32, 74% | 0.31, 79% |
| | CSPD-GP (Transformer) | 0.79, 100% | 0.78, 100% | 0.79, 100% | 0.74, 100% |
| HEPC | SigDiffusion (ours) | **0.20, 16%** | **0.18, 9%** | **0.19, 12%** | **0.21, 22%** |
| | DDO ($\gamma = 1$) | 0.25, 46% | 0.23, 38% | 0.25, 46% | 0.25, 51% |
| | Diffusion-TS | 0.85, 100% | 0.87, 100% | 0.82, 100% | 0.88, 100% |
| | CSPD-GP (RNN) | 0.52, 100% | 0.53, 100% | 0.55, 100% | 0.56, 100% |
| | CSPD-GP (Transformer) | 1.00, 100% | 1.00, 100% | 1.00, 100% | 1.00, 100% |
| Exchange rates | SigDiffusion (ours) | 0.31, 80% | 0.28, 67% | 0.28, 65% | 0.31, 74% |
| | DDO ($\gamma = 1$) | **0.24, 41%** | **0.24, 42%** | **0.25, 45%** | **0.25, 45%** |
| | Diffusion-TS | 0.72, 100% | 0.71, 100% | 0.70, 100% | 0.69, 100% |
| | CSPD-GP (RNN) | 0.59, 100% | 0.56, 100% | 0.55, 100% | 0.56, 100% |
| | CSPD-GP (Transformer) | 1.00, 100% | 0.99, 100% | 0.98, 100% | 0.99, 100% |
| Weather | SigDiffusion (ours) | 0.35, 82% | 0.34, 80% | 0.33, 76% | 0.34, 78% |
| | DDO ($\gamma = 10$) | **0.26, 45%** | **0.27, 52%** | **0.26, 46%** | **0.26, 47%** |
| | Diffusion-TS | 0.49, 100% | 0.50, 100% | 0.50, 100% | 0.49, 100% |
| | CSPD-GP (RNN) | 0.57, 100% | 0.56, 100& | 0.56, 100% | 0.56, 100% |
| | CSPD-GP (Transformer) | 0.91, 100% | 0.92, 100% | 0.91, 100% | 0.91, 100% |

We follow the score-based generative diffusion via a variance-preserving SDE paradigm proposed in Song & Ermon (2019). We tune $\overline{\beta}_{min}$ and $\overline{\beta}_{max}$ in Equation (4) to be 0.1 and 5 respectively. We use a denoising score-matching (Vincent, 2011) objective for training the score network $s_\theta$. For sampling, we discretise the probability flow ODE

$$d\mathbf{x}_t = -\frac{1}{2}\beta(t)[\mathbf{x}_t + s_\theta(t, \mathbf{x}_t)]dt, t \in [0, 1] \tag{30}$$

with an initial point $\mathbf{x}_1 \sim \mathcal{N}(0, I)$. To solve the discretised ODE, we use a `Tsit5` solver with 128 time steps. We adopt the implementation of the Predictive and Discriminative Score metrics from TimeGAN (Yoon, 2024). To satisfy the conditions for Fourier inversion, we augment the paths with additional channels as described in Theorem 3.1, and we add an extra point to the beginning of each path, making it start with 0.

The model architecture remains fixed throughout the experiments as a transformer with 4 residual layers, a hidden size of 64, and 4 attention heads. Note that other relevant works (Yuan & Qiao, 2024; Coletta et al., 2024; Biloš et al., 2023) follow a very similar or bigger architecture. We use the Adam optimiser. We run the experiments on an NVIDIA GeForce RTX 4070 Ti GPU. Table 4 details additional KS test performance metrics (see Section 5).

For the task of generating time series described in Section 5.2, we fix the number of samples to 1000, the batch size to 128, the number of epochs to 1200, and the learning rate to 0.001. The details variable across datasets are listed in Table 5. We reserve 1000 points from each dataset as an unseen test set for metric calculation.

**The mirror augmentation** For many datasets, we might not wish to assume path periodicity as required in the Fourier inversion conditions in Theorem 3.1. We observed that a useful trick in this case is to concatenate the path with a reversed version of itself before performing the additional augmentations. We denote this the *mirror augmentation*. Table 5 indicates the datasets for which this augmentation is performed.

Table 5: Datasets for long time series generation.

| Dataset | Mirror augmentation | Data points | Dimensions |
|---|---|---|---|
| Sines | Yes | 10000 | 5 |
| HEPC | No | 10242 | 1 |
| Predator-prey | No | 10000 | 2 |
| Exchange rates | No | 6588 | 8 |
| Weather | No | 10340 | 14 |

**Datasets**   As previously described in Section 5, we measure the performance of `SigDiffusions` on two synthetic (Sines and Predator-prey) and three real-world (HEPC, Exchange Rates, Weather) public datasets. We generate Sines the same way the Sine dataset is generated in TimeGAN (Yoon et al., 2019), by sampling sine curves at a random phase and frequency but changing the sampling rate to 1000. Predator-prey is a dataset consisting of sample trajectories of a two-dimensional system of ODEs adopted from Biloš et al. (2023)

$$\dot{x} = \frac{2}{3}x - \frac{2}{3}xy,$$
$$\dot{y} = xy - y. \tag{31}$$

We generate Predator-prey on a time grid of 1000 points on the interval $t \in [0, 10]$. HEPC (UCI Machine Learning Repository, 2024) is a household electricity consumption dataset collected minute-wise for 47 months from 2006 to 2010. We slice the dataset to windows of length 1000 with a stride of 200, yielding a dataset of 10242 entries. We select the *voltage* feature to generate as a univariate time series. We use the Exchange Rates dataset provided in Lai et al. (2018); Lai (2017) and slice it with a stride of 1, yielding 6588 time series. Lastly, the Weather dataset was measured and published by the Max-Planck-Institute for Biogeochemistry (Kolle, 2024). We take the first 14 features from this dataset describing the pressure, temperature, humidity, and wind conditions, and we slice the time series with a stride of 5 to get 10340 samples.

**Benchmarks**   We compare our models to four recent diffusion models for long time series generation: Diffusion-TS (Yuan & Qiao, 2024), DDO (Lim et al., 2023), and two variants of CSPD-GP (Biloš et al., 2023) - one with an RNN for a score function and one with a transformer. For Diffusion-TS and CSPD-GP, we keep the model configurations as they were proposed in the authors' implementations for datasets with similar dimensions and number of data points. One exception to this is halving the batch size for transformer-based architectures due to memory constraints. We also halve the number of epochs to preserve the proposed number of training steps. As DDO has previously only been implemented on image-shaped data, we alter the code in the authors' GitHub implementation to generate samples of shape (*time series length* x 1 x *number of channels*). We always report the performance for the RBF kernel smoothness hyperparameter $\gamma \in [0.05, 0.2, 1, 5, 10]$ corresponding to the highest predictive score. We train the model for 300 epochs (see Appendix Section J of the DDO paper by Lim et al. (2023)) with a batch size of 32 and keep the remaining hyperparameters as proposed for the *Volcano* dataset. Table 2 shows the number of parameters and computation times for each model. We use the publicly available code to run the benchmarks:

- https://github.com/morganstanley/MSML (Stanley, 2024)
- https://github.com/Y-debug-sys/Diffusion-TS (Yuan, 2024)
- https://github.com/lim0606/ddo (Lim, 2023)

**Experiments with the truncation level**   To produce Figure 4, we use a dataset generated by Equation (31), discretised to 25 points. To add non-trivial high-frequency components, we include sine and cosine waves of the first eight Fourier frequencies, with random amplitudes drawn from a standard normal distribution. To keep the model architecture consistent and avoid memory issues at higher truncation levels, we use the linearised attention mechanism from Katharopoulos et al. (2020) with a `relu` kernel, instead of the full attention mechanism. We also reduce the model's hidden size from 64 to 32.

