# OpenReview forum: "SigDiffusions: Score-Based Diffusion Models for Time Series via Log-Signature Embeddings"
_ICLR.cc/2025/Conference — ICLR 2025 Poster_

### Official Review · Reviewer_nEHh · 2024-10-29

**Soundness:** 2
**Presentation:** 1
**Contribution:** 1
**Rating:** 1
**Confidence:** 5

**Summary:**

This paper presents a new score-based diffusion modeling trick for long multivariate time series.
Their motivation is to recover the underlying dynamics of the discretized time series data via log-signature embeddings.
Here, log-signature embeddings corresponds to some algebraic structure of time underlying continuous time dynamics.
The key contribution is the derivation of a closed-form for signal recovery.
They support their theory and proposal with numerical experiments.

**Strengths:**

This work suggests a new approach to time series generation from signature inversion algorithms. The problem of interest is important. The method employed is refreshing, at least in ML.

**Weaknesses:**

## **Updated after rebuttal: Changed to strong rejection due to claiming non-original results as their own.**

---

I don't believe I'm the best person to judge this work, as I'm not well-versed in log-signature. However, I am familiar with Lie algebra, group theory, and some algebraic and topological algebra. I find this paper challenging to read and understand, even with my background. The paper feels more like a math paper than an ML paper, especially in the first half. While I appreciate the mathematics, I believe there are valuable insights here, but I suggest the authors polish the presentation to better engage the general ML community.

* Are Thm 2.1 and Thm 2.2 original results? If not, consider treating them as lemmas and specifying the reference in the title. For example, "Lemma 2.1 (Shuffle Identity, Ree 1958)."

* Poor motivation. For theory-heavy papers, motivation and intuition are important to communicate with a broader audience, including non-theorists and mathematicians. For example, in Appendix A, while the authors provide some examples, they remain very dry and don't help the reader appreciate the connection between log-signature and ML.

* As a background section, Sec. 2 is very dry and difficult to learn from, beyond memorizing definitions.

* As a methodology paper, the experiments are somewhat limited.
  -  Ablation. The paper does not include enough ablation studies. Additional ablation studies are needed to support the findings.
  - For common time series data, why only use exchange rates and weather? Why not include electricity and traffic data?

Given above, I find this paper not ready for publication. I encourage the authors to improve its accessibility for the broader ML community.

**Questions:**

Per my understanding, log signature is path integral. A few clarification questions are in order:

* Is log-signature deterministic or stochastic path?

* If it's deterministic, please justify why it's a good idea to use deterministic dynamics in stochastic diffusion process?

* How to justify the robustness of the algebraic structure against noise? Especially it's well-known time series is one of the most noisy data type.

* In `line 182`, why you describe the score function as it's not continuous? In score-based DMs, while the time is discretized for implementation purposes,  score function is continuous.

From above, I find it difficult to see the connection between DMs and log-signature embedding in Sec. 2.3.

**Details Of Ethics Concerns:**

## Updated after rebuttal:

Theorems 1 and 2 in the original submission are not original, yet the authors failed to cite the source.

I only discovered they were non-original through my clarification question: 'Are Theorems 2.1 and 2.2 original results?'

This is unacceptable. Claiming non-original work as one's own violates academic integrity. Regardless of intent, this behavior is unacceptable and should not be tolerated by the community.

---

> ### Author Response · Authors · 2024-11-25
> **Author Response**
>
> We thank the reviewer for the detailed feedback. We hope our response below will answer all raised concerns.
>
> We would first like to address the reviewer's uncertainty about the connection between diffusion models and log-signatures and provide additional intuition and motivation. To do so, we reiterate the main contributions of our work, which we believe represent significant methodological advancements in the field of time series diffusion models.
>
> ## Main contributions
>
> 1) By the Chen-Chow's Theorem, we identify the Lie algebra $\mathcal{L}^n(\mathbb R^d)$ associated to the step-$n$ nilpotent Lie group $\mathcal{G}^n(\mathbb R^d)$ as the space of log-signatures of smooth paths. The crucial consequence of this observation is that because such Lie algebra is isomorphic to a Euclidean space, we can simply leverage off-the-shelf Euclidean diffusion models on this space to generate valid log-signatures. Furthermore, because of the one-to-one correspondence between a Lie group and the associated Lie algebra, one can recover the signature in $\mathcal{G}^n(\mathbb R^d)$ from the log-signature in $\mathcal{L}^n(\mathbb R^d)$ by exponentiating. This is, to our knowledge, the first mathematically rigorous pipeline for generating variable-length time series using deterministic embeddings.
> 2) To recover the time series from its signature, we derive new closed-form inversion formulae. These exact formulae are obtained as explicit linear functionals mapping the truncated signature to the coefficients of a Fourier/polynomial basis expansion of the underlying signal; thus no training is required for this decoding. Moreover, this inversion mechanism allows to mitigate all major scalability and accuracy issues exhibited by existing signature inversion methods (see Section 5.1).
>
> ## The paper is challenging to read
> We have rewritten Section 2 in the updated paper to make it more accessible to the machine learning community. This includes additional discussion on the intuition behind the mathematical concepts and moving many technical details to the Appendix.
>
> To better clarify the connection of signatures to machine learning, we have also added discussion into Section 5, clarifying that the main aim of the experiments, which is to show that (log)signatures, combined with our closed-form inversion, provide a particularly effective embedding for time series diffusion models.
>
> We recognise that signatures are inherently theoretical and may remain challenging to grasp. We kindly ask the reviewer to review the revised sections and let us know if they have any further suggestions.
>
> ## Thm 2.1 and Thm 2.2
> These theorems are not original results. We have now changed them to lemmas.
>
> ## Ablation studies
> We would like to clarify that, in addition to weather and exchange rate data, we already use electricity data (see the HEPC dataset), as the reviewer suggested. While we agree that testing on a broader range of datasets would be ideal, our work is conducted within the constraints of an academic setting with limited computational resources.
>
> Additionally, we would like to draw the reviewer's attention to Figure 8 and Appendix D, where we provide a detailed discussion on the quality of our inversion method across different bases. We have also added a new discussion on the choice of the signature truncation level in Appendix E.
>
> If the reviewer has specific ablation experiments they believe are missing from our paper, we would greatly appreciate their suggestions.
>
> ## Answers to questions
>
> - **Is the log-signature a deterministic or stochastic path?** The (log)signature is a deterministic transformation (embedding) of a path into a Lie algebra. Since this Lie algebra is a linear space, it allows us to perform diffusion directly on it, as described in our main contributions.
> - **Why use deterministic dynamics in stochastic diffusion process?** Beyond the many desirable properties that signatures possess for machine learning (see Section 2.2), using this deterministic transformation is both fast and exact for computation and inversion. As demonstrated by our performance metrics, it retains most of the critical information about the shape of multivariate time series, making it an effective embedding for diffusion processes.
> - **Robustness of the signatures against noise** Signatures provide a global representation of a path rather than focusing on individual point values, making them inherently robust to noise. The iterated integrals computed during signature calculation act as filters, smoothing out noise in the process.
> - **In line 182, why do you describe the score function as it’s not continuous?** On line 182, we are referring to the continuous generalisation of noise in a diffusion model governed by stochastic differential equations, as opposed to discrete noise values used in DDPM. We have clarified the wording in the updated paper. Please let us know if further explanation is required.

---

> > ### Comment · Reviewer_nEHh · 2024-11-25
> > **Rejection Due to Academic Integrity Concerns**
> >
> > Thank you for your response and revisions. I have reviewed them carefully.
> >
> > While the draft is clearer now, some parts still require further polishing.
> >
> > However, **I must advocate for a strong rejection of this paper.**
> >
> > The reason is straightforward—**treating non-original work as your own crosses the boundary of academic integrity.** Regardless of intent, this is unacceptable, and I believe the community should not tolerate it either.
> >
> > Please take this as a serious lesson and ensure it does not happen again.

---

> ### Author Response · Authors · 2024-11-25
> **Author Response**
>
> We strongly disagree with the reviewer's claims.
> In the previous version of the paper, we have referenced both of these theorems. Specifically, the text immediately before stating the theorems was as follows:
> - **Thm 2.1 (Shuffle Identity):** "This result is originally due to Ree (1958). For a modern proof see (Cass & Salvi, 2024, Theorem 1.3.10)."
> - **Thm 2.2 (Chen-Chow):** "It is established by the following fundamental result due to Chen (1957; 1958), which can also be viewed as a consequence of Chow’s results in Chow (1939)."
>
> These two theorems are classical and celebrated results in rough path theory, as everybody knows in the community, so it is of course not in our intention to claim ownership. Our expectation is that readers read the surrounding text to a statement, not just the statement itself.

---

> ### Comment · Reviewer_nEHh · 2024-11-25
>
> Let's agree to disagree. I prefer to leave the final decision to the chairs and ethics reviewers.
>
> Since you’ve decided to push back, please excuse my direct but truthful response.
>
> To clarify my position for future readers (as they may not have access to the original submission), the original wording was, at best, ambiguous.
>
> For example, consider the following quoted statements:
>
> > This result is originally due to Ree (1958). For a modern proof see (Cass & Salvi, 2024, Theorem 1.3.10).
>
> > It is established by the following fundamental result due to Chen (1957; 1958), which can also be viewed as a consequence of Chow’s results in Chow (1939).
>
> Why not use a straightforward statement like 'from [original source], we have the following...'?
>
> What does 'originally due to' mean? Typically, this phrase is used when the author extends or builds upon the original result. Similarly, what does 'it is established by' mean? This phrase usually refers to a proof or derivation.
>
> **Why use such implicative language?**
>
> **It is the authors' responsibility to ensure the clarity and accuracy of their statements. Providing ambiguous phrasing that skirts the boundaries of proper attribution and then claiming innocence is unacceptable.**
>
> Regardless of the original intention—whether an honest mistake due to poor language choice or not—I believe the research community should not tolerate such practices at any level.
>
> I encourage the authors to reflect on this feedback and prioritize clarity and proper citation in future work.
>
> --
>
> To add on, whether the theorems are celebrated results in rough path theory or not is not essential. This is a Machine Learning conference.

---

### Official Review · Reviewer_U13T · 2024-11-01

**Soundness:** 3
**Presentation:** 2
**Contribution:** 3
**Rating:** 6
**Confidence:** 3

**Summary:**

This paper proposes an algorithm for generating particularly long multi-series time series. The proposed method, SigDiffusion, embeds time series into a Lie algebra and is a diffusion model constructed from the perspective of Lie algebra structure.

**Strengths:**

- This paper proposes a new method for constructing a time series generation model for multiple series. The model method is score-based diffusion, but it is a method that is theoretically robust because it is constructed based on Lie algebra analysis.
- The proposed method is also very useful for objects with long series lengths.
- It is constructed with theoretical proof, so I think it will be useful in terms of leading to a wide range of future expansion. Personally, I am very interested in the possibility of applying it to texts and leading to the construction of an LLM.

**Weaknesses:**

This paper is strong in the theoretical part, and that is important, but on the other hand, the description of the experimental part is poor. It is not clear what the experiment proves in relation to the paper's claim. It is unclear whether it is showing that it is excellent as a generative model for time series, or whether it is showing that it is excellent in score-based diffusion, but since it is being compared with TS-Diffusion, I think it is showing that it is excellent in the overall generative model for multiple series. Since time series generation models are being actively developed, it is necessary to compare the most advanced methods, and it should be explained that appropriate methods have been selected for comparison. There are so many different new methods, and the evaluation metrics and data sets are all different, that it is impossible to judge whether they are producing sufficient results.
Also, the Discriminative Score is generally a better indicator when the value is larger, and this can be recognized even by reading the explanation, but it seems that a smaller value is better. I think it would be better to clarify the definition. It seems that the predictive score is defined in the same way as the loss used in training, but the loss in training does not always match the loss in prediction performance, so the definition should be clearly stated.

**Questions:**

- What is the claim you want to prove in Table 1, and after looking again at the most advanced methods, please tell us why the comparison you have chosen is appropriate as a way of proving your claim. There may be cases where you change the comparison. Table 1 in [1] will be useful for a list of cutting-edge methods.

[1] Y. Yang et al., A Survey on Diffusion Models for Time Series and Spatio-Temporal Data, arXiv:2404.18886

- Please clarify the definitions of the evaluation indicators in Table 1.
If the above points are clarified, I will raise the score.

---

> ### Author Response · Authors · 2024-11-25
> **Author Response**
>
> We thank the reviewer for the detailed feedback. We hope our response below will answer all raised concerns.
>
> To address the reviewer’s concerns regarding the claims in our paper, we would first like to reiterate its main contributions, which we believe represent significant methodological advancements in the field of time series diffusion models.
>
> ## Main contributions
>
> 1)  By the Chen-Chow's Theorem, we identify the Lie algebra $\mathcal{L}^n(\mathbb R^d)$ associated to the step-$n$ nilpotent Lie group $\mathcal{G}^n(\mathbb R^d)$ as the space of log-signatures of smooth paths. The crucial consequence of this observation is that because such Lie algebra is isomorphic to a Euclidean space, we can simply leverage off-the-shelf Euclidean diffusion models on this space to generate valid log-signatures. Furthermore, because of the one-to-one correspondence between a Lie group and the associated Lie algebra, one can recover the signature in $\mathcal{G}^n(\mathbb R^d)$ from the log-signature in $\mathcal{L}^n(\mathbb R^d)$ by exponentiating. This is, to our knowledge, the first mathematically rigorous pipeline for generating variable-length time series using deterministic embeddings.
> 2) To recover the time series from its signature, we derive new closed-form inversion formulae. These exact formulae are obtained as explicit linear functionals mapping the truncated signature to the coefficients of a Fourier/polynomial basis expansion of the underlying signal; thus no training is required for this decoding. Moreover, this inversion mechanism allows to mitigate all major scalability and accuracy issues exhibited by existing signature inversion methods (see Section 5.1).
>
> Given these contributions, the claim we aim to substantiate is that (log)signatures, combined with our closed-form inversion, provide a particularly effective embedding for time series diffusion models. Signatures capture the global structure of a time series and are agnostic to the number of encoded time steps. This property allows SigDiffusions to generate paths with arbitrarily fine-grained discretization at no additional computational cost.
>
> ## Choice of benchmarks
> We benchmark against the current state-of-the-art diffusion models explicitly designed to be robust to time series length and fine-grained discretisation. We have added a discussion justifying each of our chosen benchmarks to Section 5.
>
> We have reviewed Table 1 from the survey provided by the reviewer, considering each DDPM or score-based model listed in the multivariate time series category for generation tasks:
> - **[1], [5], and [6]** are application-focused papers that propose generating time series in specific domains using simple architectures for the denoising model, such as U-Net or LSTM. These works do not aim to develop state-of-the-art models for general time series generation, nor do they introduce methodological advancements in this area. There is no evidence to suggest that these approaches would generalise better to a wide range of datasets compared to the state-of-the-art diffusion baselines we have already evaluated.
> - **[7] and [8]** propose methods for speech enhancement, i.e., obtaining a clean voice recording from a noisy one via diffusion models. While the models include a generative step for noise removal from real-world samples, they do not generate unseen time series.
> - Similarly to [7],  **[3]**  directly uses real data during the generation process rather than learning an underlying distribution of time series.
> - **[2]** employs an ODE-based approach to encode and decode time series into latent embeddings, which are then fed into a diffusion model. This results in processing and generating time series in a point-by-point manner, rather than treating the time series as a global object, as in our method. Therefore, this work belongs to a parallel research stream of Neural-ODE methods mentioned in Section 4 and is not directly comparable with our approach.
> - As we understand, **[4]** was designed for and tested on generating one-dimensional time series only.
> - **[9]** makes diffusion models more robust to noisy samples. While it conducts experiments on generating time series, it does not claim to offer a novel approach to time series generation, nor does it provide any benchmarking against other models in this area.
>
> Based on this analysis, we hope it is clear that none of the models listed offer a suitable baseline for benchmarking against our method. We are happy to answer any additional questions.
>
> We have mentioned several of the reviewer's suggested benchmarks in Section 4.
>
> ## Discriminative and predictive score
> We have now clarified the definitions in Section 5 of the updated paper.
>
> [1]arxiv.org/abs/2303.12281
> [2]arxiv.org/abs/2311.03303
> [3]arxiv.org/abs/2312.05550
> [4]arxiv.org/abs/2312.07981
> [5]ieeexplore.ieee.org/document/10353969
> [6]arxiv.org/abs/2308.09857
> [7]arxiv.org/abs/2212.11851
> [8]arxiv.org/abs/2302.14748
> [9]arxiv.org/abs/2402.02081

---

> > ### Author Response · Authors · 2024-11-29
> > **Author Response**
> >
> > We thank the reviewer very much for their valuable feedback and for deciding to raise the score.

---

> ### Comment · Reviewer_U13T · 2024-11-25
>
> Since the validity of the proposed method and the experiment have been clarified to a certain extent, I will raise the score.

---

### Official Review · Reviewer_ry7B · 2024-11-03

**Soundness:** 3
**Presentation:** 3
**Contribution:** 3
**Rating:** 6
**Confidence:** 3

**Summary:**

Authors propose a generative model for multivariate time-series data, SigDiffusion. The diffusion processes generates samples from a distribution in a (latent truncated) log-signature space (coming from Rough Path theory), which serves as an efficient embedding of a time series. The authors discuss improved methods for recover from the log-signature to time-series path space — namely they propose some new closed-form inversion formulas to recover the time-series trajectories from the log-signature as polynomial series.

The two main contributions are:
1. Application of score-based generative diffusion model to space of truncated log-signatures, which is
2. Improvements to signature inversion algorithm going from the log-signature space to trajectory space.
    1. This is done by path as expressed as a series in different polynomial bases, which can be shown to converge.

These two claims are backed up by empirical experiments:
- For the signature inversion: synthetic evaluation using example paths random sine waves + gaussian noise,
- For the diffusion model application time-series: Two synthetic data (sines, Lotka–Volterra system) and three real data: Household Electric Power Consumption, Exchange Rates, and Weather.

**Strengths:**

- The paper is clear, well organized, and well written (see later comments on section 2)
- The overall scheme using log-signature is grounded in sound mathematical theory from rough path theory and signature method literature. The claims in Theorems 3.1 and 3.2 seem reasonable (though I did not check in high detail) and appear to be new contributions that address a known problem in literature.
- Efficacy of doing diffusion in log-signature space backed up by empirical evidence
    - Per Table 2. The speedup by doing the generative process in the truncated log-signature is an advantage with comparable or better performance to other diffusion method for continuous-time paths.

**Weaknesses:**

### Contribution, Theory, and Experiments:

- Doing the diffusion in log-signature (or the space of any integral transform/transformed feature or embedding) is a relatively straightforward idea (similar ideas are discussed in related literature of section 4). Primary contribution seems to me to the closed form signature inversion formula, which then enables one to do the diffusion processes in the log-signature space. I am more or less convinced that the presented polynomial series would approximate the original signal. But all numerical illustrations done in this aspect are only with synthetic data, and not using real data (though the synthetic illustrations are quite extensive, they seem to mostly cover of the case of a low frequency signal + noise).
    - Since the signature inversion is a primary claim, it may be useful to provide some illustrations using the real datasets about reconstruction quality (e.g., in terms of $\ell_2$ error akin to Fig 8.) from log signature (e.g., number of basis, type of basis, properties of the data analogous to Appendix D.)
    - The runtime to recover using the closed form inversion formula in terms of number of basis and number of truncated signatures is not explicitly stated.
- It may be useful to have at least a comparison to one non-diffusion baed generative time series model (i.e., pick one from section 4, following from TimeGAN paper).

### Organization and Writing:

- Disclaimer: I not an expert in signature method or rough path theory:
    - Not coming from this background, I think presentation in full generality of Section 2 is, in my opinion, not very accessible.
    - The present development may be more suitable for an appendix. The exposition in provided references (e.g., Fermanian et al. 2023) was more clear to me.

### Errata:
- Line 213: acronym ST is not defined, assume it refers to (log)-Signature Transform.

**Questions:**

- Could authors provide some results on signal recovery for real signals? (e.g. some signals have non-noise high frequency characteristics that may not be well captured without higher order terms of polynomial series, you could back this with up with some synthetic signals as well).
- In Appendix E of experimental details could authors provide some discussion about which inversion scheme (and how many basis authors used) authors chose for the real data and why?
- Why is the order of the log-signature order chosen to be 4 in your experiment? Since the runtime growth is highly unfavorable in the order $\mathcal{O}[(d^{n+1}-1)/(d-1)]$ and mildly unfavorable in the dimension of the signal.
- On the runtime aspect, to make work self contained, could authors provide some discussion (extending Figure 1) about how much runtime is required to transform between the truncated log-signature space (extending discussion in lines 198-205), and go from the log-signature space to the path space using Theorems in 3.1 and 3.2 (i.e., extending Table 2) and comment on runtime as compared to previous method (e.g., Inversion method).
- Connection to long sequences specifically is unclear to me, could you elaborate on why SigDiffusion is more suitable for long sequences?

Happy to adjust my score if these concerns are addressed

---

> ### Author Response · Authors · 2024-11-25
> **Author Response**
>
> We thank the reviewer for the detailed feedback. We hope our response below will answer all raised concerns.
>
> ## Signature inversion illustration for real datasets
> As requested, we have added counterparts of Figure 8 and Figure 10 for two of our real-world time series datasets, showcasing the inversion quality. You can find them in the Appendix of the newly uploaded paper version as Figures 9 and 11.
>
> We would like to clarify that to retrieve the first $N$ basis coefficients, one needs a signature truncated at level $N + 2$. Since the signature inversion formulae are exact, the inversion quality for a given signature depends only on how well the underlying signal is approximated by the retrieved collection of $N$ basis coefficients (see Appendix D for discussion).
>
> In their first question, the reviewer correctly pointed out that some signals have non-noise high-frequency characteristics that may not be well captured without higher-order terms of polynomial series. In such cases, a sufficiently high truncation level of the signature is required to capture these details. This can, in turn, increase the complexity of the diffusion task. We have added a paragraph discussing this trade-off to Appendix E.
>
> ## Runtime of inversion
> See Section 3.3 of the updated paper, where we have added a discussion about the time complexity of our inversion.
>
> ## Comparison to a non-diffusion benchmark
> We have included the performance of TimeGAN on the Sines and Predator-prey datasets in Table 1 and Table 2. Due to the increasingly long training times listed in Table 2, we plan to add the metrics for the other datasets in the coming days.
>
> ## Presentation of Section 2
> We have rewritten Section 2 in the updated version of paper to make it more accessible to the machine learning community. This includes additional discussion on the intuition behind the mathematical concepts and moving many technical details to the Appendix.
>
> We recognize that signatures are inherently theoretical and may remain challenging to grasp. We kindly ask the reviewer to review the revised section and let us know if further clarifications or improvements would be helpful.
>
> ## Acronym ST is not defined
> We have fixed this in the updated version.
>
> ## Answers to questions
>
> - **Could authors provide some discussion about which inversion scheme (and how many basis coefficients) they chose for the real data and why?**
> For all datasets, we used the Fourier inversion scheme with step-$4$ log-signatures, allowing us to recover the Fourier coefficients up to $a_2$ and $b_2$. This choice was guided by initial cross-validation. Since the focus of this work is to demonstrate the effectiveness and robustness of (log)signatures as time series embeddings in diffusion models, we kept the experimental setup the same for all datasets.
> To supplement this discussion, we have also added plots visualising sample quality as a function of the truncation level of the signature in Appendix E (see Figure 13).
>
> - **Could authors provide some discussion (extending Figure 1) about how much runtime is required to transform between the truncated log-signature space, and go from the log-signature space to the path space using (i.e., extending Table 2) and comment
> on runtime as compared to previous methods.**
> The time complexity of our inversion formulae is listed in Section 3.3. The computational complexity of computing a signature is $O(LNd^N)$, where $L$ is the time series length, $N$ is the truncation level, and $d$ is the number of dimensions in a path. Lemma A.6 in [2] shows that converting between signatures and log-signature requires $N\sum_{k=1}^N kd^k$ multiplications. Following derivations similar to A.1.3. of [3] we see that this conversion has a time complexity of $O(N^2d^N)$.
> In the context of Table 2, these operations are all exact and deterministic, taking only seconds per dataset to compute. Hence, their runtime is negligible compared to model training.
> Lastly, the time complexity of the Insertion method is not stated by the authors, but [4] provides an empirical analysis of the runtime. The Optimisation method is learning-based, so it is not possible to reason about time complexity.
>
>
> - **Connection to long sequences specifically is unclear.**
>     The signature transform encodes the global shape of a time series rather than processing individual points, meaning the embedding complexity is independent of the number of encoded time steps. This enables SigDiffusions to generate paths with any discretisation, from irregularly sampled paths to continuous functions. This is a significant advantage over models using learned latent embeddings for time series representation. We have added discussion to Section 5 about how each of our chosen benchmarks deals with this problem as well.
>
> [1] https://arxiv.org/pdf/2210.02040
> [2]https://discovery.ucl.ac.uk/id/eprint/10156498/2/Shujian_Liao_PhD_Thesis.pdf
> [3] https://openreview.net/pdf?id=lqU2cs3Zca
> [4] https://arxiv.org/pdf/2304.01862

---

> ### Comment · Reviewer_ry7B · 2024-11-25
>
> I want to thank the authors for their detailed and thorough responses. I have reviewed carefully the responses and questions to the other reviewers as well.
>
> This work has two contributions:
> 1. Signature inversion algorithm going from the log-signature space to trajectory space via expression into a polynomial basis.
> 2. Application of score-based generative diffusion model to space of truncated log-signatures
>
> I believe that contribution (1) is well supported, valid, and useful contribution to the literature.
>
> However application aspect (2) is a less clear contribution. Having reviewed the content of the updated figures (thanks for providing them), I think that the core issue is still the difficulty with data of high frequency, and with noise (e.g., you may imagine some path which is governed by a stochastic differential equation), with the number of bases chosen, the method would more or less only ignore those high frequency components/noise.  To some extent, this is okay for *representation* (which is what the truncated log-signature may be interpreted to be. though of course you want to capture the high frequency component, and some statistical aspect of the noise), however as a *generative* model your goal is to produce samples which are close to the true data distribution, for which these components are very important. If say the signal has high frequency noise, then you need to generate a trajectory which has frequency noise (say for example it comes from some realization of a stochastic differential equation, like the price of a security). This type of data is quite typical in the time-series setting. So, I am not convinced but this from the presented results. How to do this in the *truncated*-log signature space without incurring significant algorithmic penalty not entirely clear to me. This is a fundamental limitation of the model, so I think this should not be relegated to appendix. I think that it would be prudent to also some sampled generated by your model, in contrast to some true trajectories, as well as some trajectories generated by the other models.
>
> Regarding the writing: updated section two is much more accessible, I think this is better. I also agree that the a rigorous formulation is important, and should not be sacrificed (so it's good to put in appendix).
>
> Based on the above, for now I shall keep my current recommendation.
>
> As for the claim of another reviewer on academic misconduct: Based on the evidence presented to me, I agree with the authors that those background theorems were clearly referenced, as part of the development of the rough path theory. Furthermore, the fact that these results are part of the development and are not to be construed as original should be abundantly clear to any ordinarily skilled addressee. I have conducted this review on the basis that the misconduct claim is unsubstantiated, but I shall defer the final decision to the chairs and ethics reviewers.

---

> ### Author Response · Authors · 2024-11-26
> **Author response**
>
> We sincerely thank the reviewer for their thorough and thoughtful review.
>
> We agree that high-frequency signal components present a limitation of our approach. We have now added a section to the main body of the paper (see Section 6) discussing this.
>
> However, we respectfully disagree with the reviewer’s doubts regarding our contributions to the field of generative modeling. Accurately modeling high-frequency noise in multivariate time series remains a significant challenge and, to our knowledge, an open problem for all state-of-the-art models.
>
> While truncated log-signatures do smooth out high-frequency noise, they have many other desirable properties for time series modeling. Specifically, they are highly effective at capturing the overall shape and cross-dimensional dependencies in multivariate time series, and they are robust to various discretization schemes, including irregular sampling and continuous functions. Additionally, using log-signatures rather than learned latent embeddings greatly simplifies the learning task, making SigDiffusions particularly efficient (see Table 2).
>
> To better illustrate the sample quality of SigDiffusions in comparison to other benchmarks, we have added t-SNE visualizations (see Figure 13) and sample trajectory plots (see Figure 14).

---

> > ### Comment · Reviewer_ry7B · 2024-11-26
> >
> > Thanks for the quick turnaround and the new results, and yeah I agree that generative modeling for time-series as a field generally needs a lot of work.
> >
> > As for “Capturing the overall shape and cross-dimensional dependencies in multivariate time series, and they are robust to various discretization schemes, including irregular sampling and continuous functions” are all great benefits, and I am convinced by this, and the log-signature transform is indeed fast. I think this is a good contribution to the representation of time series.
> >
> > Overall I think this is a good work with solid merits, solid contributions, and a very solid theoretical grounding. However, I still stand by stand by my opinion on the *generative modeling* of time-series. Generated samples should resemble the empirical distribution (in the aspects you mention, and in “generation” of noise and regular high frequency patterns). I believe this area as a whole, is not there yet, there’s a lot of work to be done.
> >
> > Still I think that the narrative surrounding those actual merits, and limitations, of the signature transform method are not sufficiently presented clearly enough in both the experiments and the presentation of the manuscript. If these are highlighted clearly in the manuscript as the problem being addressed then I think this I would be solid work (and why I hold off on raising the score). I encourage you to elaborate on these aspects, especially that ability to capture overall shape and cross-dimensional dependencies in multivariate time series (e.g., following figure 14), My worry about making the generative model a big part of the narrative is that you have a good generative model for time series, but only for the low frequency component of the signal.

---

> ### Author Response · Authors · 2024-11-29
> **Author Response**
>
> As suggested by the reviewer, we have revised the narrative of Section 5.2 in the Experiment section to emphasize the trade-off between the fidelity of time series representation and model capacity. Additionally, we have explicitly stated that our diffusion model experiments focus on highlighting the ability to model the shape and cross-channel dependencies of time series using signatures with low truncation levels.
>
> In the final paper version, we propose adding another figure to Section 5.2.1, similar to Figure 4, but showcasing samples from models trained on signatures of progressively higher truncation levels. This figure will illustrate that as the truncation level increases, the generated samples become more oscillatory but deviate further from the true shape. We did not have enough time to produce this visualization for the current submission.
>
> We have also added a discussion to Section 6 about how this limitation could be addressed in future work.
>
> While further revisions to the paper are not possible anymore, we are eager to keep discussing smaler re-writes to the narrative of the paper in the discussion forum and implementing them in the final version of the paper.

---

### Meta-Review · Area_Chair_Ew3D · 2024-12-13

**Metareview:**

The paper introduces SigDiffusion, a generative algorithm for creating long multivariate time series. It utilizes a Lie algebra-based approach, embedding time series in a log-signature space derived from Rough Path theory. SigDiffusion generates samples in this latent space and provides improved closed-form inversion formulas to reconstruct time-series trajectories as polynomial series from the log-signature representation.

We have a lot of discussions about this paper. In general, the reviewers appreciate the work and would like to support its publication. The main issue of the paper is that the reviewer had flagged an ethics concern for this work due to the authors claiming non-original results (Theorem 1 and Theorem 2 of the original submission) as their own. However, after taking a close look, we see that the authors clearly cited the sources for the two theorems. For not causing any confusion, we would like the authors to revise the wording from "theorem" to "lemma" or "proposition" for clarity, and also cite them properly. Please also incorporate all the discussions with the reviewers and revise the final version of the paper.

**Additional Comments On Reviewer Discussion:**

In general, the reviewers appreciate the work and would like to support its publication. The main issue of the paper is that the reviewer had flagged an ethics concern for this work due to the authors claiming non-original results (Theorem 1 and Theorem 2 of the original submission) as their own. However, after taking a close look, we see that the authors clearly cited the sources for the two theorems.

---

### Decision · Program_Chairs · 2025-01-22

Accept (Poster)